# Identifying *s*-wave pairing symmetry in single-layer FeSe from topologically trivial edge states

Zhongxu Wei [1,2,10], Shengshan Qin[3,4,5,10], Cui Ding[2,6], Xianxin Wu [7], Jiangping Hu [4,5,8], Yu-Jie Sun [1,9] ✉, Lili Wang [2] ✉ & Qi-Kun Xue[1,2,6,9] ✉

Determining the pairing symmetry of single-layer FeSe on $SrTiO_3$ is the key to understanding the enhanced pairing mechanism. It also guides the search for superconductors with high transition temperatures. Despite considerable efforts, it remains controversial whether the symmetry is the sign-preserving *s*- or the sign-changing $s_\pm$-wave. Here, we investigate the pairing symmetry of single-layer FeSe from a topological point of view. Using low-temperature scanning tunneling microscopy/spectroscopy, we systematically characterize the superconducting states at edges and corners of single-layer FeSe. The tunneling spectra collected at edges and corners show a full energy gap and a substantial dip, respectively, suggesting the absence of topologically non-trivial edge and corner modes. According to our theoretical calculations, these spectroscopic features can be considered as strong evidence for the sign-preserving *s*-wave pairing in single-layer FeSe.

Heavily electron-doped iron chalcogenides $A_xFe_2Se_2$[1,2] (A = alkali metal), $(Li_{1-x}Fe_x)OHFeSe$[3,4], and single-layer FeSe on $SrTiO_3$[5,6] have attracted significant attention due to their high transition temperatures ($T_c$) and simple Fermi surface that consisted only of electron pockets centered at the M point of the Brillouin zone (BZ)[7–9]. Identifying the pairing symmetry of these materials is crucial to reveal the pairing mechanism fully. Previous angle-resolved photoemission spectroscopy[7–9] and scanning tunneling microscopy/spectroscopy (STM/S) measurements[10–12] have consistently demonstrated anisotropic gap without nodes, suggesting nodeless *d*-, sign-changing $s_\pm$- and sign-preserving *s*-wave states as competitive candidates (Fig. 1a). The nodeless *d*-wave state is unlikely because the strength of spin-orbit coupling (SOC) between electron pockets is comparable to that of pairing[13–15]. Moreover, the

behavior of Caroli–de Gennes–Matricon states in the vortex core disfavors *d*-wave pairing[16,17]. On the other hand, the controversy over the two leading contenders, i.e., sign-preserving *s*- and sign-changing $s_\pm$-wave, remains unresolved, despite extensive impurity-scattering investigations[18–24] including measurements employing the recently developed quasiparticle interference technique[18–20,25–27]. The conflict stems from the difficulties in proving the practical magnetic or nonmagnetic nature of the impurities under investigation[21] and distinguishing bound states under weak scattering potential[28]. In addition, although phase-sensitive tests based on tunneling junctions have been proposed much earlier[29], hitherto no experimental result has been reported for iron chalcogenides. Overall, the pairing nature of these materials is still an open issue and more deterministic characterization is desired.

[1]Department of Physics, Southern University of Science and Technology, Shenzhen 518055, China. [2]State Key Laboratory of Low-Dimensional Quantum Physics, Department of Physics, Tsinghua University, Beijing 100084, China. [3]School of Physics, Beijing Institute of Technology, Beijing 100081, China. [4]Kavli Institute of Theoretical Sciences, University of Chinese Academy of Sciences, Beijing 100049, China. [5]CAS Center for Excellence in Topological Quantum Computation, University of Chinese Academy of Sciences, Beijing 100049, China. [6]Beijing Academy of Quantum Information Sciences, Beijing 100193, China. [7]CAS Key Laboratory of Theoretical Physics, Institute of Theoretical Physics, Chinese Academy of Sciences, Beijing 100190, China. [8]Beijing National Research Center for Condensed Matter Physics, and Institute of Physics, Chinese Academy of Sciences, Beijing 100190, China. [9]Quantum Science Center of Guangdong–Hong Kong–Macao Greater Bay Area (Guangdong), Shenzhen 518045, China. [10]These authors contributed equally: Zhongxu Wei, Shengshan Qin. ✉e-mail: sunyj@sustech.edu.cn; liliwang@mail.tsinghua.edu.cn; xueqk@sustech.edu.cn

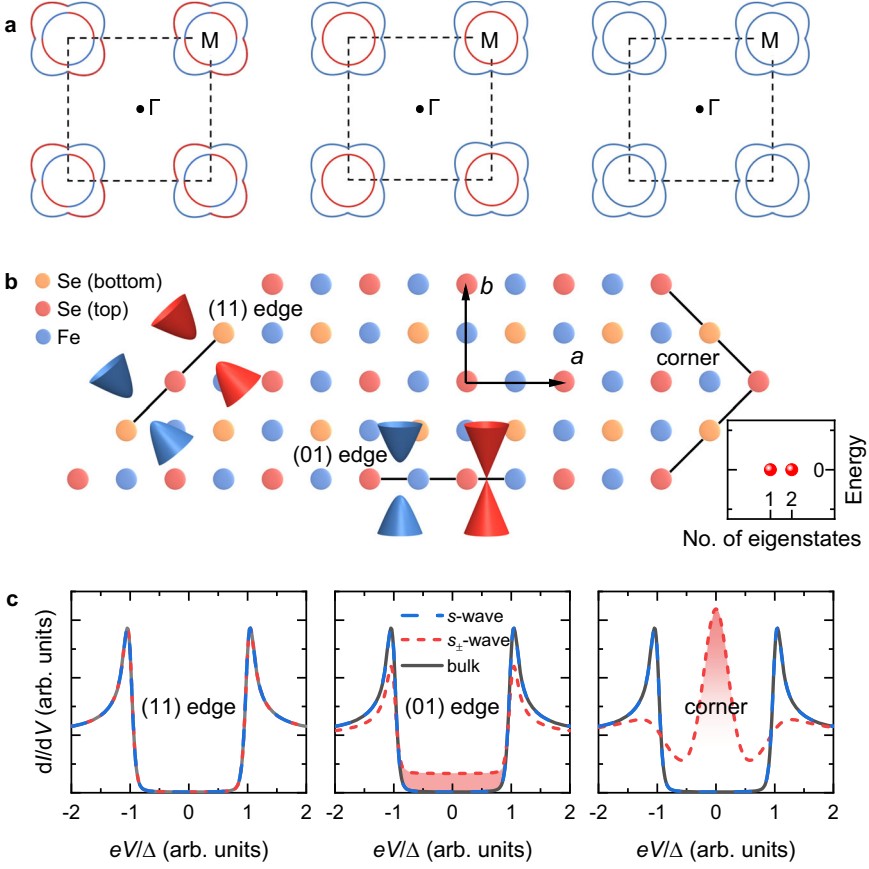

**Fig. 1 | Edge and corner modes of single-layer FeSe. a** Schematic illustration of the Fermi surface of single-layer FeSe consisting of two electron pockets centered at the M point of the Brillouin zone (black dashed lines). Different colors denote the opposite sign of order parameters. The left, middle, and right panels depict nodeless $d$-, sign-changing $s_\pm$-, and sign-preserving $s$-wave pairings with band hybridization taken into account, respectively. **b** Definition of edges and the corner. The blue and red cones indicate the band structures at various edges in the case of $s$- and $s_\pm$-wave pairing, respectively. The lower right panel indicates zero-energy modes for the $s_\pm$-wave case. **c** Supposed tunneling spectra at the (11) and (01) edges and the corner for the $s$- (blue) and $s_\pm$-wave (red) cases. The gray lines are spectra away from the edge/corner. The main contributions of the Dirac mode and the Majorana mode to the tunneling spectrum are highlighted in red shading. The horizontal axis is normalized by the superconducting energy gap $\Delta$.

Recently, we have developed a promising way from a topological point of view to settle the long-lasting debate between the $s$- and $s_\pm$-wave pairing of heavily electron-doped iron chalcogenides[30]. Our theoretical work[30] shows that the topological properties of an iron-based superconductor with a centrosymmetric space group P4/nmm are only sensitive to the pairing signs of the two electron pockets centered at the M point. Specifically, as shown in Fig. 1b, the inversion center in Se−Fe−Se triple-layer is at the nearest Fe−Fe bond center rather than the Fe site, which naturally generates the Rashba-type SOC between the next-nearest-neighbor Fe sites[15,30–33]. Accompanied by the unique lattice structure and SOC, anomalous band degeneracies along the BZ boundary develop. As a consequence, a sign-changing $s_\pm$-wave pairing leads to a second-order topological superconducting state which supports two Dirac cones at the (01) edge and a pair of Majorana zero-energy modes at the corner between the (11) and ($1\bar{1}$) edges (Fig. 1b)[30]. In contrast, sign-preserving $s$-wave states remain topologically trivial even in the presence of inversion symmetric Rashba SOC. Therefore, the gapless edge modes and zero-energy corner modes, which can be directly probed by STM/S (Fig. 1c), serve as the smoking-gun evidence to distinguish the $s$- and $s_\pm$-wave pairing.

Here we investigate the superconducting states at isolated edges and corners of single-layer FeSe, and then discuss the pairing symmetry from the topological perspective. Our spectroscopic investigations demonstrate full gap along both (01) and (11) edges though the superconductivity gets suppressed with moving to the isolated edges. Furthermore, there is no evidence for corner Majorana modes at the

intersection between the (11) and ($1\bar{1}$) edges. Combined with our theoretical calculation[30], these topologically trivial superconducting states support the sign-preserving $s$-wave pairing in single-layer FeSe.

## Results

Figure 2a shows a typical topographic image of single-layer FeSe on SrTiO₃(001) with a sharp edge along the (11) direction, judged from the inset zoom-in atomically resolved image. The dark contrast in the upper right area is contributed by the exposed substrate. The apparent height of the single-layer FeSe relative to the SrTiO₃ surface is ~660 pm at a sample bias ($V_s$) of 1 V, slightly larger than the out-of-plane lattice constant of FeSe (550 pm) due to the difference in density of states[34]. As displayed in Fig. 2b, two sets of tunneling spectra far from (red) and near (blue) the edge along the colored arrows in Fig. 2a do not show significant differences in the zero-bias conductance. Figure 2c, d presents the spatial distribution of the superconducting gap $\Delta$ (half the distance between coherence peaks) and the zero-bias conductance extracted from the mapped d$I$/d$V$ curves over the area labeled by the black dashed box in Fig. 2a. Obviously, $\Delta$'s near the (11) edge are smaller than those in bulk, which may be due to the lattice discontinuity. The superconducting gap extracted by a temperature smeared Bardeen−Cooper−Schrieffer (BCS) density of states gives consistent results (Supplementary Fig. 1). Nevertheless, the zero-bias conductance, our main concern, is uniform in real space (Fig. 2d). Excluding anomalies induced by local defects marked by white boxes (Fig. 2a, c, d), both the raw and normalized column-averaged zero-bias

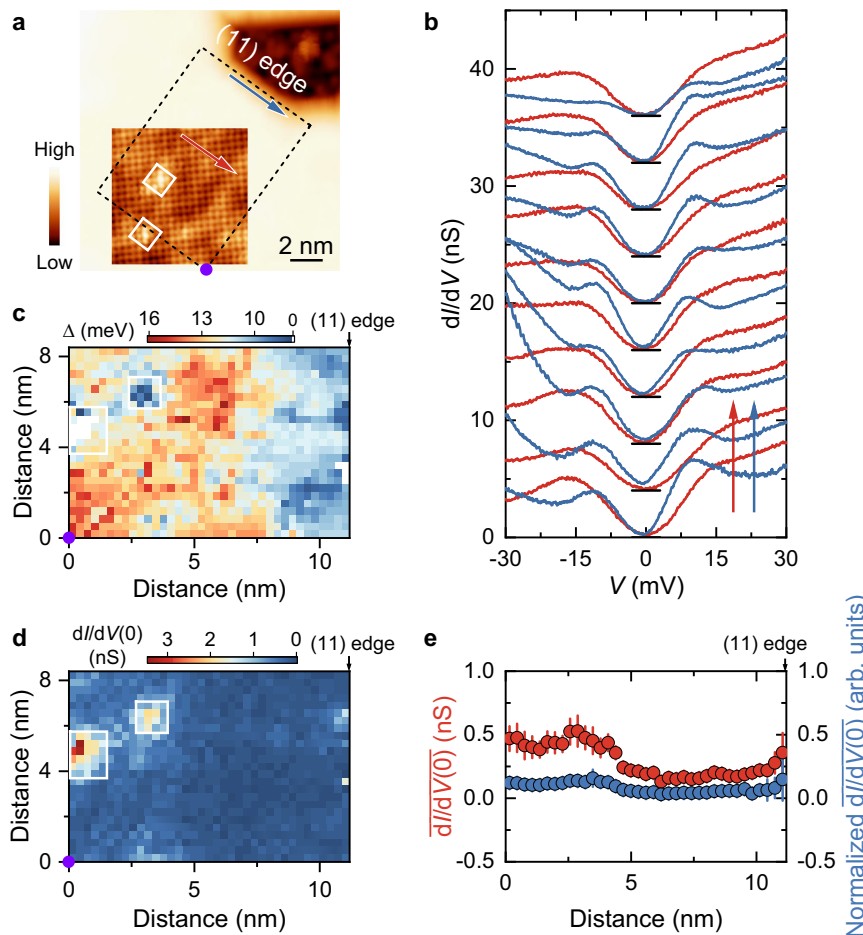

**Fig. 2 | Electronic states at the (11) edge of single-layer FeSe. a** STM topographic image ($V_s = 1\,V$, $I_t = 50\,pA$) of the (11) edge. Inset: atomically resolved image ($V_s = 50\,mV$, $I_t = 500\,pA$) of FeSe. **b** Two sets of tunneling spectra measured along the blue and red arrows in (**a**). The data are vertically shifted in the direction of the corresponding colored arrows for clarity. The black lines indicate individual zero conductance. **c, d** Energy gap map (**c**) and zero-bias conductance (d$I$/d$V$(0)) map (**d**) obtained from the spectroscopic mapping over the area outlined by the black dashed box in (**a**). The purple circles mark the origin point. The white boxes in (**a**), (**c**), and (**d**) mark local defects and the anomalies in $\Delta$ and d$I$/d$V$(0) induced by these defects, respectively. **e** Raw and normalized column-averaged zero-bias conductance as a function of the distance to the (11) edge. The normalization method is described in Supplementary Note 1. The error bar refers to the standard error and its value has been doubled for clarity.

conductance shown in Fig. 2e are almost insensitive to the distance to the (11) edge. The absence of finite constant conductance across the superconducting gap meets the expectation along (11) edges, as depicted in the left panel in Fig. 1c, providing a reference for analyzing the related results at the (01) edge.

Figure 3a depicts an STM image of (01) edge. The inset zoom-in atomically resolved image shows ordered lattices without any impurities and defects, except for the commonly observed weak electronic contrast[35,36]. The ordered lattice extends to the (01) edge, indicating preserved lattice symmetry near the edge. In addition, this specular edge extends more than 17 nm (Supplementary Fig. 3), about six-fold of the coherence length of single-layer FeSe[18], acting as an excellent platform of edge modes as proposed in ref. 30 (see Supplementary Note 2 for details). Consistent with the observation along the (11) edge, the coherence peak shrinks near the (01) edge while the zero-bias conductance remains unchanged, as resolved from the tunneling spectra shown in Fig. 3b. For further verification, we collect a spectroscopy map over the whole region shown in Fig. 3a. Figure 3c and Supplementary Fig. 1 plot the spatial distribution of $\Delta$ derived from the coherence peaks spacing and the temperature smeared BCS density of states, respectively, showing a decreasing evolution similar to that observed near the (11) edge. In case neglect any trace of the edge modes, we have checked all tunneling spectra near the (01) edge but

only found fully gapped Fermi surface. As shown in Fig. 3d, the zero-bias conductance map has no anomalies. Furthermore, both the raw and normalized column-averaged values keep almost unchanged with moving to the (01) edge, as plotted in Fig. 3e. The minor zero-bias conductance difference in Figs. 2e and 3e could be due to varied tunneling contribution from the substrate (related to the coverage and distribution of FeSe films) and tip apexes. We repeatedly get the same results at (01) edges (see Supplementary Note 3 for more data). The results above give the conclusion that the superconducting states at the (01) edge of single-layer FeSe are topologically trivial.

To further check the topological properties of single-layer FeSe, the corner states are studied. Figure 4a shows a topographic image containing a corner formed by the (11) and (1$\bar{1}$) edges, verified by the Se lattice shown in Fig. 4b. Due to the epitaxial growth of FeSe, intersected (11) and (1$\bar{1}$) edges are rarely observed and generally extend only a few nanometers. Fortunately, despite the relatively small corner size, the topological properties can still be judged from the featured a pair of in-gap bound states with opposite energies induced by the hybridization of the supposed corner modes with the neighboring topological modes (see ref. 37 and Supplementary Note 2 for details). Figure 4d presents a set of tunneling spectra measured along the gray arrow in Fig. 4a, with the top two spectra taken on the SrTiO$_3$(001) surface. Though the superconducting gap is gradually suppressed upon approaching the

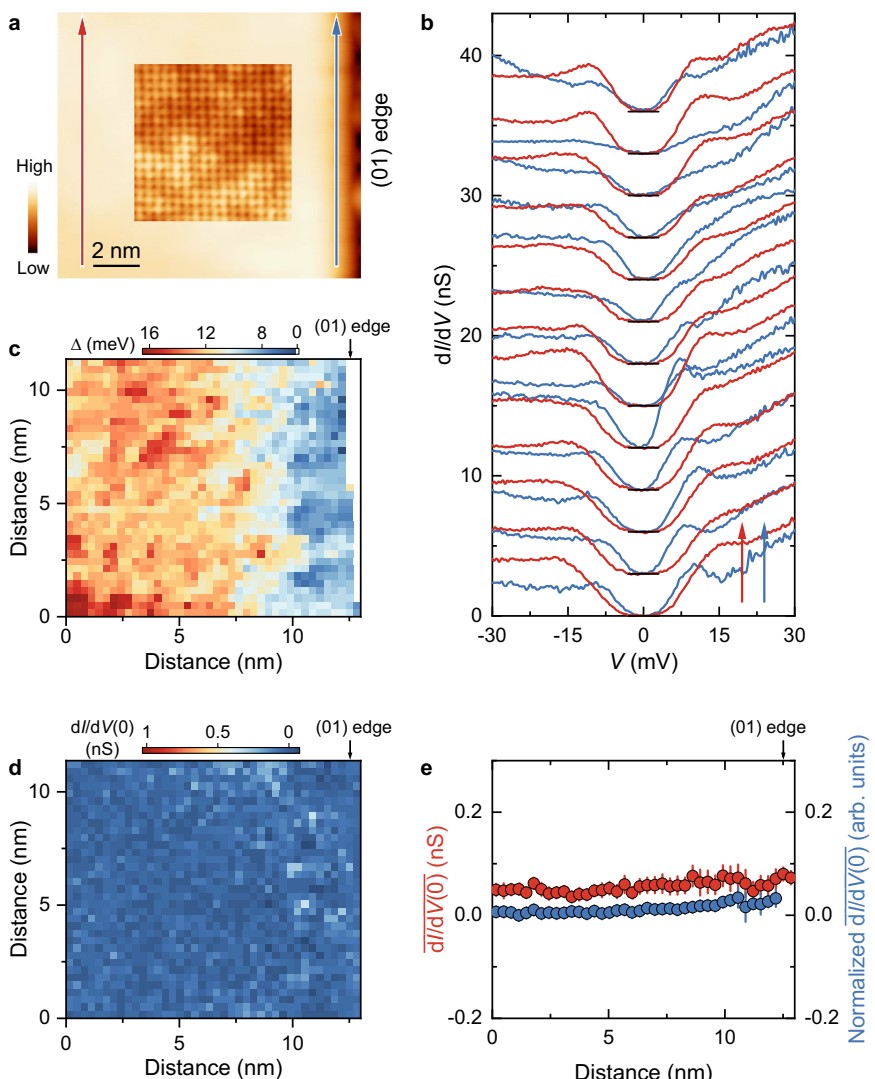

**Fig. 3 | Electronic states at the (01) edge of single-layer FeSe. a** STM topographic image ($V_s = 1$ V, $I_t = 50$ pA) of the (01) edge. Inset: atomically resolved image ($V_s = 50$ mV, $I_t = 500$ pA) of FeSe. **b** Two sets of tunneling spectra measured along the blue and red arrows in (**a**). The data are vertically shifted in the direction of the corresponding colored arrows for clarity. **c, d** Energy gap map (**c**) and zero-bias conductance (d$I$/d$V$(0)) map (**d**) obtained from the spectroscopic mapping over the same field of view as in (**a**). **e** Raw and normalized column-averaged zero-bias conductance as a function of the distance to the (01) edge. The normalization method is described in Supplementary Note 1. The error bar refers to the standard error and its value has been doubled for clarity.

corner, there is no discernable anomalies within the superconducting gap. That is, zero-bias conductance peak (ZBCP) and in-gap bound states as supposed for $s_\pm$-pairing are missing. For careful verification, we perform d$I$/d$V$ mapping over the area outlined by the black box in Fig. 4a containing the intersection between the (11) and (1$\bar{1}$) edges, and find relatively uniform zero-bias conductance as exemplified in Fig. 4c. It should be emphasized that the results for such corners are highly reproducible (see Supplementary Note 4 for more data), confirming the topologically trivial superconductivity in single-layer FeSe.

Before concluding the pairing symmetry of single-layer FeSe, we discuss possible actual effects on edge/corner modes beyond the theoretical model[30]. First issue is the SrTiO$_3$ substrate and its coupling with FeSe. The non-uniform strain arising from the structural instability of SrTiO$_3$[38,39] may introduce local mirror symmetry breaking, which is detrimental to the topological superconductivity. Nevertheless, the local mirror symmetry breaking, if any, is relatively weak, as confirmed by topographic images (Figs. 2a, 3a, 4a and Supplementary Figs. 3, 5–8). In addition, even if the two Dirac edge modes at the (01) edge could hybridize and be gapped out under weak mirror symmetry breaking, the single Majorana Kramers' pair located at the

isolated corner can exist stably[40]. On the other hand, the SrTiO$_3$(001) surface contributes no electronic states around the Fermi energy (Fig. 4d and Supplementary Figs. 5–6), indicating that the edge/corner modes will exponentially decay in the substrate and have to be localized around the edge/corner. We have performed numerical simulations and found that the topological states under sign-changing $s_\pm$-wave pairing still firmly exist though weakened even on metallic substrate (see Supplementary Note 5 for details). The second issue is the suppressed pairing and superconductivity around the edge/corner. The decreased gap magnitude near the edge/corner implies reduced $T_c$. The estimated reduced $T_c \sim 33$ K at the edge, under the assumption of constant $2\Delta / k_B T_c$ ratio for single-layer FeSe, is still much larger than the experimental temperature of 4.8 K. In this situation, no anomaly is expected on the zero-bias conductance (see Supplementary Note 6 for details). Furthermore, we introduce a position-dependent superconducting gap into the numerical simulations and demonstrate that the topological edge/corner modes stably exist, as long as the system is in the sign-changing $s_\pm$-wave pairing state (see Supplementary Note 6 for details). As experimental evidence supporting the above discussion, the spectra collected under

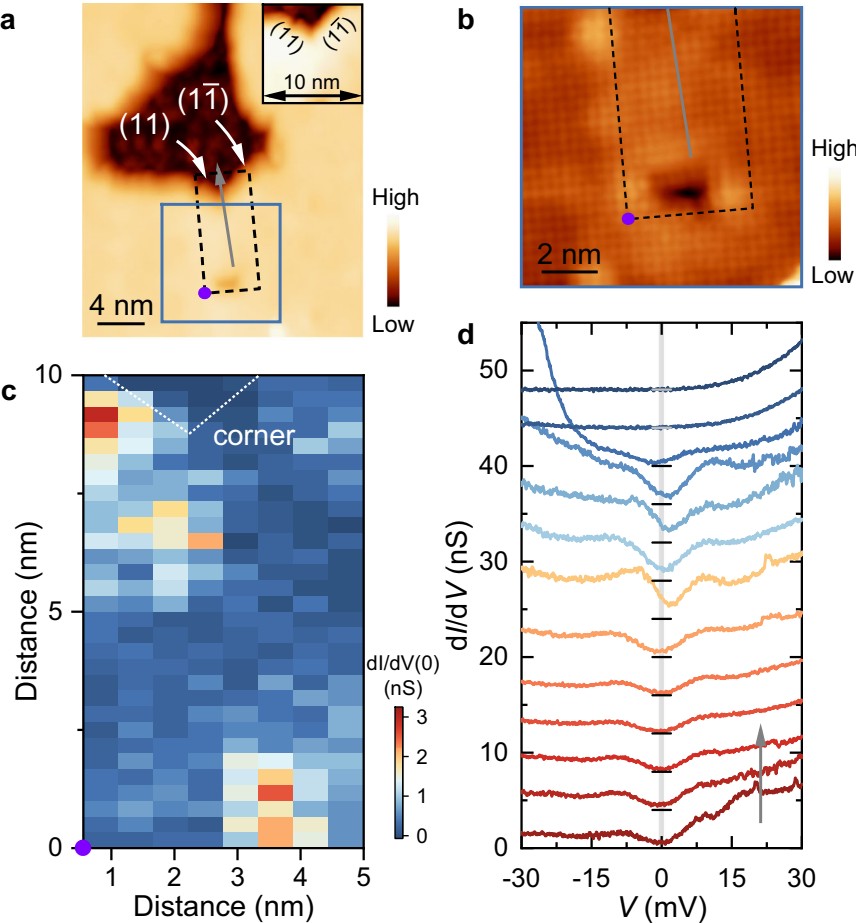

**Fig. 4 | Topological properties of the corner between the (11) and ($1\bar{1}$) edges.**
**a** STM topographic image ($V_s = 1\,V$, $I_t = 50\,pA$) of the corner. Inset: zoom-in image. **b** Atomically resolved image ($V_s = 50\,mV$, $I_t = 500\,pA$) taken from the area outlined by the blue box in (**a**). **c** Zero-bias conductance map obtained from the spectroscopic mapping over the area indicated by the black dashed box in (**a**). The purple circles mark the origin point. **d** The tunneling spectra collected along the gray arrow in (**a**). The data are vertically shifted in the direction of the gray arrow for clarity. The black and gray horizontal lines indicate the position of the zero conductance of corresponding tunneling spectra. The gray shaded line is a guide to the evolution of the zero-bias conductance.

$I_t / V_s = 10\,nS$ (see "Methods" for details) with our equipment can capture anomalies in differential conductance of meV-scaled energy (refs. 21,41,42). A pertinent example is the successful capture of impurity states near the (11) edge (Fig. 2d and Supplementary Note 7). Therefore, the spectra measured at the edge/corner intrinsically reflect the topological property of single-layer FeSe.

The topological property of single-layer FeSe provides essential information on its pairing symmetry[30]. In the case of $s_\pm$-wave pairing, the second-order topological superconductivity arises in centrosymmetric single-layer FeSe with the help of additional glide-plane and mirror symmetries[30]. Specifically, two Dirac cones and one single Majorana Kramers' pair are expected, respectively, at the (01) edge and the corner between the (11) and ($1\bar{1}$) edges. The Dirac cones contribute finite energy-independent density of states within the superconducting gap, and the Majorana Kramers' pair leads to a quantized ZBCP in the tunneling spectrum. In the case of $s$-wave pairing, however, the superconducting states at all edges and corners are topologically trivial, i.e., fully gapped. Therefore, as summarized in Table 1, studying the spectroscopic features at edges/corners is a practical way to distinguish between the sign-preserving $s$- and sign-changing $s_\pm$-wave states. The tunneling spectra shown in Figs. 2–4 definitively exclude the existence of edge modes and Majorana modes. Consequently, we unambiguously conclude that the pairing symmetry of single-layer FeSe is the sign-preserving $s$-wave rather than the sign-changing $s_\pm$-wave.

It is worth noting that previous works, refs. 16,43, have also investigated edge states of single-layer FeSe. Reference 16 reports the fully gapped spectra along two kinds of (01) edges, i.e., step edge between 1 unit cell (uc) and 2 uc FeSe, and stacking edge in 1 uc FeSe along SrTiO$_3$ step. In the former case, the superconducting bottom layer of the 2 uc FeSe side[44] smoothly extends to the 1 uc FeSe side, indicating that such configuration is actually the edge of the non-superconducting upper layer of the 2 uc FeSe. In the latter case, FeSe on adjacent SrTiO$_3$ terraces is non-separate, since the thickness of FeSe (550 pm) is larger than the step height of SrTiO$_3$ (390 pm). The physics near such edge is elusive, and whether there is a response in electronic states to the topological superconductivity needs more detailed study. Under a simple assumption that FeSe films on both sides are weakly linked, the tunneling spectra at the edge are expected to be fully gapped, regardless of the pairing symmetry. Here, all edges and corners investigated in our work are constructed of 1 uc FeSe and vacuum, which are ideal for detecting the edge/corner modes. Reference 43 reports a pair of emergent conductance peaks located near the superconducting gap at the (01) edge.

**Table 1 | Features of tunneling spectra of single-layer FeSe at various edges and corners**

|  | (11) edge | (01) edge | (11) & ($1\bar{1}$) corner |
|---|---|---|---|
| $s$-wave | Gapped | Gapped | Gapped |
| $s_\pm$-wave | Gapped | Gapless | ZBCP |

However, the conductance peak, which can only be resolved after a normalization done by subtracting the spectrum far from the edge, is most likely due to the reduction in superconducting gap near the edge and therefore are not related to topological properties (see Supplementary Notes 8 and 9 for more details).

## Discussion

In conclusion, we fabricate high-quality single-layer FeSe by molecular beam epitaxy and investigate the electronic structures at different edges and corners by STM/S. We neither observe gapless edge modes at the (01) edges nor detect ZBCP at the corners between the (11) and (1$\bar{1}$) edges. The topologically trivial superconducting states are solid evidence for the sign-preserving s-wave pairing symmetry of single-layer FeSe. Our achievements also pave a promising way to determine the pairing symmetry of other iron-based superconductors such as single-layer Fe(Se,Te)[45], $K_xFe_2Se_2$[46], and $(Li_{1-x}Fe_x)OHFeSe$. More delicate experiments are to be designed to identify the pairing glue supporting s-wave state such as orbital fluctuation, phonon, etc.[47–49]. Finally, we would like to point out that the applicability of our method may be affected if the material has long-range magnetic order. Nevertheless, currently there is no conclusive evidence for long-range magnetic order in single-layer FeSe.

## Methods

### Sample preparation

High-quality FeSe thin films were grown on Nb-doped $SrTiO_3(001)$ (0.05 wt%) substrates by molecular beam epitaxy. The surface is $TiO_2$ terminated after heating to 1150 °C for 15 min. Then FeSe films were fabricated by co-evaporating Fe (99.995%) and Se (99.999%) from standard Knudsen cells onto $SrTiO_3$ kept at 450 °C. The growth rate was -0.024 uc per minute. By controlling the coverage to be less than 1 uc, well-defined edges with various orientations form between FeSe and vacuum. After annealing at 480–490 °C for 4.5 h, the sample was transferred in situ to the STM chamber.

### STM experiments

Topographic images were obtained by the constant-current mode. The tunneling spectra were measured using a standard lock-in technique with a sample bias ($V_s$) of 50 mV, a tunneling current ($I_t$) of 500 pA, and a bias modulation of 0.5 mV at 937.2 Hz. The differential conductance $dI/dV(V)$ of all spectra is calibrated proportionally by scaling $dI/dV(V_s)$ to $I_t/V_s$. Commercial Pt/Ir tips were calibrated on Ag films before performing STM/S experiments. All STM/S experiments were carried out at 4.8 K.

### Numerical simulations

A toy model is used to simulate the edge and corner modes in various conditions. The model Hamiltonian reads as

$$H_0(\mathbf{k}) = \left[2t\left(\cos k_x + \cos k_y\right) - \mu\right]s_0\sigma_0 - 2R\left(\sin k_x s_2 + \sin k_y s_1\right)\sigma_3 + 4t'\cos\frac{k_x}{2}\cos\frac{k_y}{2}s_0\sigma_1,$$

where $s_i$ and $\sigma_i$ are the Pauli matrices standing for the electron spin and two Fe sublattices, respectively. $k_x$ and $k_y$ are defined according to the primitive lattice translations along the next-nearest-neighbor Fe–Fe directions. In $H_0(\mathbf{k})$, $\mu$ is the chemical potential, $t$ ($t'$) is the hopping between the next-nearest-neighbor (nearest-neighbor) Fe lattice sites, and $R$ is the Rashba-type spin-orbit coupling between the next-nearest-neighbor Fe lattice sites which arises from the mismatch between the Fe lattice sites and the inversion center.

One can use the above toy model to simulate the topological superconductivity of single-layer $FeSe/SrTiO_3$ in the sign-changing $s_\pm$-wave pairing state due to the following facts: (i) The single-layer $FeSe/SrTiO_3$ merely has two electron pockets centered at the M point. (ii) For an electronic system with spin-orbit coupling, the space group P4/nmm respected by FeSe only has one irreducible representation,

i.e., more specifically one 4D irreducible representation, at the M point. Namely, at the M point all the energy bands are 4-fold degenerate in FeSe and all the bands respect the same low-energy effective model. $H_0(\mathbf{k})$ is just the lattice version of the low-energy effective model. (iii) The model described by $H_0(\mathbf{k})$ captures the key topological properties of the sign-changing $s_\pm$-wave pairing state in FeSe correctly (see Supplementary Note 5 for details).

## Data availability

All data needed to evaluate the conclusions in the study are present in the paper and/or the Supplementary Information. The data that support the findings of this study are available from the corresponding authors upon request.

## Code availability

The computer code used for numerical simulations and theoretical understanding is available upon request from the corresponding authors.

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

## Acknowledgements

This work is supported by the National Natural Science Foundation of China (Grants No. 12074210, 52388201, 12141402, 11790311 and No. 12204222) and the National Basic Research Program of China (Grant No. 2017YFA0303303).

## Author contributions

Z.X.W., C.D., and L.L.W. synthesized the single-layer FeSe films. Z.X.W. and C.D. performed the scanning tunneling microscopy/spectroscopy measurements. S.S.Q., X.X.W., and J.P.H. performed the numerical simulations and provided theoretical understanding. Z.X.W., S.S.Q., and L.L.W. wrote the manuscript. All the authors discussed and contributed to the manuscript. Y.-J.S., L.L.W., and Q.-K.X. conceived the project.

## Competing interests

The authors declare no competing interests.
