## [Peer Review File · Nature Communications]

Identifying s-wave pairing symmetry in single-layer FeSe from topologically trivial edge statesREVIEWER COMMENTS

Reviewer #1 (Remarks to the Author):

This paper builds on previous conceptual insight won by authors in a paper (ref 30) where they analyze the consequences of being able to define a local electric dipole by using local sub-symmetries of a space group that does not show, overall, any reason for showing electric dipolar behavior. The idea is interesting, yet it remains to be tested. As a matter of fact, the references given in addition to ref 30, do not describe a purely electric (and non-magnetic) effect associated to the said sub-symmetries. Thus, while there is certainly something interesting in these ideas, it is not clear to me that it can be driven so far in the consequences as to propose that this will firmly favor the appearance of a delicate Majorana mode.

In the present paper, authors analyze the tunneling conductance of single layer FeSe, which is in itself a feat. Nevertheless, authors present atomic resolution studies that are only partial, not reaching the truly interesting edge behavior. Generally, the result is that when approaching a step authors observe a full gap instead of states within the gap. However, the measurements are not atomically resolved. Atomic resolution stops quite far from the edges. Furthermore, the full gap might also be due to changes in the band structure close to the step. Instead of changes in the in gap conductance, authors observe significant changes in the gap magnitude. This suggests that the critical temperature is decreasing, which produces very unfavorable conditions to obtain the bound states proposed in the reference 30.

Unfortunately, I am not convinced that authors have sufficient evidence to support their claim of absence of $s\pm$ superconductivity in this system. Furthermore, writing of the paper is such that it seems that authors claim that $s\pm$ superconductivity is absent in many other iron based systems too, which is quite unjustified. Unless a very large additional set of data is provided, showing clearly what happens to the band structure in this system in presence of edges, I do not think that this paper can be published. In re-writing, authors might consider all possible explanations for their results and state more clearly what is understood and favored by their data and what should be studied on later work. In addition, I do not feel that the data present the expected degree of novelty.

To aid authors in re-writing and submit somewhere else to a more specialized venue, I would suggest to consider the following aspects:

- Authors should specify clearly the applicability of the approach of ref 30. While this is in principle quite generic, here it is applied to single layer FeSe and it is not clear to me how this would influence other iron based materials. Authors should be much more careful in specifying that their results refer to FeSe single layers.
- Authors should better explain what a "second order superconducting state" means. And, in particular, why such a "second order state" would be better than the effects that are being studied using the "first order state". The argument that impurities produce scattering bound states with an inherent complexity and that instead symmetry based arguments could help out is very difficult to hold. One needs to create a bound state also when considering symmetry arguments (actually, the usual bound states can be traced down to symmetry arguments too). The interaction of the edge with the rest of the system does play a role in the creation of a bound state. To me, there is no fundamental difference when it comes to creating in-gap modes.
- Authors state that "obviously, Δ 's near the (11) edge are smaller than those in the bulk, which may be due to lattice discontinuity. Nevertheless, the zero-bias conductance, our main concern, is uniform in real space." I completely disagree with such an approach. The zero bias conductance is very much connected to the rest of the tunneling conductance in the superconducting phase. Changing T_c does lead to changes that can easily preclude the appearance of in-gap modes. Along this line, the change in T_c is not considered when authors discuss the role of the substrate in the detection of edge states.
- Authors present "anomalies induced by local defects." But what is the origin of the in-gap modes created by such local defects if it is not some sort of gap-varying behavior? There is one defect where the gap is completely suppressed (left white box of Fig 2c).
- Along the line of thought of previous point, authors might argue that these local defects are magnetic. Is then magnetism completely absent from edges? This is very difficult to understand.
- In Figs 2 e and 3e, authors should plot the normalized tunneling conductance, not the absolute value. This would allow the reader to understand how close that is to "zero".
- It is quite disturbing and time consuming to see that authors chose red to represent the

tunneling conductance at the edge, and the same color for the size of the gap far from the edge.
- Authors use a finite Dynes term, which is just between 100 and 200 microeV large. This is less than one per cent of the gap magnitude. It can probably be removed without changing too much the rest of the parameters.

- I also disagree with the discussion about Reference 7. There, it is shown that one has a topological edge state and superconductivity. This shows that there are relevant band structure changes at the edge and that these cannot be neglected and should be incorporated in the discussion about what happens at the edge. What is the influence of such changes in the "second order superconducting state"? Clearly, there is an influence, because T_c changes and I do not understand why in such conditions one should expect a "second order" effect to hold.

Reviewer #2 (Remarks to the Author):

In the manuscript by Z. Wei et al. submitted to Nature Communications, the authors investigate the superconducting pairing symmetry in monolayer FeSe grown on STO using scanning tunneling spectroscopy (STS) measurements. Many groups have investigated the enhanced superconducting properties of monolayer FeSe on STO by various techniques but conclusive evidence for the pairing symmetry has been elusive. Being able to resolve this issue will help further our understanding of the enhanced superconducting state as well as superconductivity in the iron pnictides in general and hence suitable for Nature Communications. Taking advantage of the FeSe structure and the location of the inversion center in the monolayer, the authors show how different pairing symmetries give rise, or not, to topologically protected spectral features (i.e. Majorana modes) at specific domain edges and corners. While confirmation of zero bias peaks being Majorana modes is a tricky task for topological superconductors, here the authors simply look for the existence of a zero bias peak and the lack of one can be connected to a specific s-wave pairing symmetry. Reproducibility is key for such an investigation and the authors have revealed their results are consistent across different edges and corners on the sample. Hence the authors use topology to discriminate between the different pairing symmetries with high confidence. The approach is sound, and the results are indeed interesting. I recommend the manuscript for publication in Nature Communications in its current form.

Reviewer #3 (Remarks to the Author):

The authors use an STM technique on single-layer FeSe on a SrTiO_3 substrate with the eventual goal to discern between two superconducting pairing candidates: an s_{\pm} -wave or an s_{++} -wave state where the pairing sign on the two Fermi surfaces is changed or preserved, respectively. The particular intent is based on the work by parts of the authors in Ref. 30 [S. Qin et al., Phys. Rev. X 12, 011030 (2022)], where they predicted the s_{\pm} -wave state to be a second-order topological SC state with Dirac cones and Majorana states on particular edges and corners—while the sign-preserving s_{++} -wave state is non-topological. Using STM the authors study various edges and corners of their sample. They do not observe any signatures of the Dirac cones or Majorana states, leading to their conclusion that the pairing state should be the non-topological sign-preserving s_{++} -wave state.

The paper is written in a comprehensible way, and the results are clearly presented and communicated. Furthermore, the results contribute to the understanding of the pairing symmetry in this highly-debated iron-based SC, and they are of interest for a broader audience. If the authors can convincingly answer my questions, I would recommend a publication in Nature Communications.

1. From a view into Ref. 30 and Eq. (8) therein, I understand that as soon as there is a sign change in the s-wave SC order parameter components on the two Fermi surfaces at the M point, the state is inevitably topologically non-trivial—regardless any other system parameters—, is that correct? (If so, one could maybe emphasize that in the paper.)

2. What is the length scale that determines if the geometrical extension (edge, corner) is large enough to be identified as such by the superconductor ? And more specific, how certain are you that the corner you studied with an extension $\sim 4\text{nm}$ is large enough for a Majorana state to emerge there ?

3. With respect to your numerical study and Fig. S7; in the case with $V_{\{b\}}=6$ [Fig. $f_{\{1,2\}}$ and $g_{\{1,2\}}$] the increase of the inter-layer hopping decreases the weight of the Majorana corner state. Yet, with $V_{\{b\}}=5$ [Fig. $f_{\{3,4\}}$ and $g_{\{3,4\}}$] it seems to me that the Majorana state benefits from the increased inter-layer hopping. Any explanation on that ?

Author Rebuttals to Comments

We thank the reviewers for their constructive criticisms, comments, and approvals. We have taken these comments and suggestions very seriously and devoted great efforts to performing new experiments, numerical simulations, and analysis in order to address the reviewers' concerns. With the intensive revisions, we hope the reviewers will be convinced about the novelty, completeness, and reliability of our manuscript.

Below, please find our response letter and the list of main changes. In the point-by-point response letter, our replies are in blue typeface. The modifications to the manuscript/Supplementary Information are described or quoted in red. A detailed list of changes is attached at the end of this document.

-----

Response to the Reviewer #1

-----

This paper builds on previous conceptual insight won by authors in a paper (ref 30) where they
analyze the consequences of being able to define a local electric dipole by using local sub-
symmetries of a space group that does not show, overall, any reason for showing electric dipolar
behavior. The idea is interesting, yet it remains to be tested. As a matter of fact, the references given
in addition to ref 30, do not describe a purely electric (and non-magnetic) effect associated to the
said sub-symmetries. Thus, while there is certainly something interesting in these ideas, it is not
clear to me that it can be driven so far in the consequences as to propose that this will firmly favor
the appearance of a delicate Majorana mode.

**Authors:** We thank the reviewer for the detailed review of our work and the positive comments,
“The idea is interesting”. Considering the comments and questions raised by the reviewer in the
following, we would like to address and clarify the key points in our present work before presenting
a detailed one-to-one response.

**Figure R1 | Typical Fermi surfaces for the iron pnictides (a) and iron chalcogenides (b).** Notice
that the topological superconducting state in Ref. [30] is sensitive to the phase difference between
the pairing order on the α and β Fermi surfaces, but has nothing to do with the γ Fermi surface.

(i) *In the present work, we merely focus on the single-layer FeSe/SrTiO₃ system.* As pointed out
by the reviewer, our work builds on the theoretical work in Ref. [30]. Ref. [30] has demonstrated
that if the *s*-wave pairing has opposite pairing signs on the two electron Fermi pockets centered at
the M point of the Brillouin zone (BZ), the iron chalcogenides will be in the topological
superconducting state and the topological edge/corner states are expected as illustrated in Figure 1

in the main text. Here, it is worth pointing out that the iron-based superconductors have two families,
*i.e.*, the iron pnictides and the iron chalcogenides; And the main difference between these two kinds
of materials is that, the iron chalcogenides have no (or very small) Fermi pockets at the BZ center
(the Γ point), while there are multiple Fermi pockets centered at the Γ point in the iron pnictides, as
sketched in Figure R1. According to the theoretical work, the topological superconducting state can
and only can help to detect the pairing signs on the two electron Fermi pockets at the M point, *i.e.*,
the pairing phase on the α and β Fermi surfaces in both the iron chalcogenides and the iron pnictides
(though the superconducting state with sign-changed s_{\pm} -wave pairing on the α and β Fermi surfaces
is merely predicted in the iron chalcogenides). The limitation of the topological superconducting
state is that it cannot provide information on the pairing phase on the γ Fermi pockets at the Γ point
(Figure R1a). Since the single-layer FeSe/SrTiO₃ only has two electron pockets centered at the M
point and high-quality samples with high-quality edges can be easily prepared, it is reasonable to
say that single-layer FeSe/SrTiO₃ is an ideal platform to apply the theory. In the future, we would
like to characterize the superconducting states at isolated edges and corners in other iron
chalcogenides.

(ii) We agree with the reviewer that carrying out experiments is the best way to test a theory. One
may suggest verifying the validity of our theoretical predictions in other iron-based superconductors
whose order parameter is known to be the sign-changed s_{\pm} -wave as shown in Figure 1 in the
manuscript. However, we want to point out that *no iron-based superconductor (or other*
*superconductors with similar lattice and electronic structures required in Ref. [30]) has been*
*undoubtedly confirmed to possess such sign-changed s_{\pm} -wave pairing currently*. For example,
(Li_{1-x}Fe_x)OHFeSe is a well-studied superconductor whose Fermi surface consists only of two
electron pockets centered at the M point, but its pairing symmetry remains controversial despite
significant efforts [e.g., Phys. Rev. B **94**, 134502 (2016); Nat. Phys. **14**, 134 (2018); Phys. Rev. B
**98**, 134503 (2018)]. In fact, this is exactly the starting point for developing the theory in Ref. [30],
because it can provide new strong evidence to distinguish the superconducting pairing symmetry in
these systems.

(iii) We want to emphasize that the theory in Ref. [30] is solid and general. The topological property
of the system only depends on the lattice structure (the P4/nmm space group), the electronic
structure (the Fermi pockets on the BZ boundary), and the pairing structure (the sign-changed s_{\pm} -

wave pairing mentioned in the above). For the iron-based superconductor, the lattice structure (we
specify it to the P4/nmm symmetric system) and the electronic structure are fixed. Therefore, the
topological property is solely determined by the pairing signs on the two Fermi pockets at the M
point. Besides, we want to point out that the topological superconductivity from the sign-changed
s_{\pm} -wave state is not only verified by a minimal lattice model, but also has been confirmed by a more
realistic model with all the five d orbitals from Fe taken into account (Appendix G in Ref. [30]).
According to the bulk-edge and bulk-corner correspondence, the topological invariant in the sign-
changed s_{\pm} -wave state, *i.e.*, the even mirror winding number (details in Ref. [30]), can result in the
topological edge and corner modes incontrovertibly. In addition, we can find some similarity
between our topological superconductivity and the famous topological crystalline insulating state in
SnTe. In our case, the topological superconductivity is protected by a mirror-symmetry-protected
winding number which is 2; in SnTe, the topological insulating state is protected by a mirror-
symmetry-protected Chern number which is 2. The topological state in SnTe is expected to host two
surface Dirac cones on the surface respecting the mirror symmetry and one helical hinge modes at
the hinge respecting the mirror symmetry [Nat. Commun. **3**, 982 (2012); Sci. Adv. **4**, eaat0346
(2018)], which is quite similar to the condition in Ref. [30]. The mirror-symmetry-protected surface
Dirac cones in SnTe have been observed [Nat. Phys. **8**, 800 (2012)], whereas the helical hinge modes
remain to be detected experimentally. In principle, the corner zero-energy modes in our case can be
detected more easily since they contribute much larger local density of states compared to the helical
hinge modes in SnTe.

(iv) As pointed out by the reviewer, the topological superconducting state in Ref. [30] is closely
related to the local electric dipole arising from the special lattice structure (*i.e.*, the space group). In
fact, such local electric dipole has also been studied in other theoretical works [e.g., Nat. Phys. **10**,
387 (2014); Phys. Rev. Lett. **119**, 267001 (2017); arXiv: 2204.02449], and its effects (the sublattice-
distinguished spin polarizations on the energy bands stemming from the Rashba-type spin-orbit
coupling induced by such local electric dipole) have been observed experimentally, such as the
study in $\text{LaO}_{0.55}\text{F}_{0.45}\text{BiS}_2$ which respects the same space group P4/nmm [Nat. Commun. **8**, 1919
(2017)].

Besides the above points, we understand the reviewer's concerns raised in the following, such as
the effect of the suppressed superconducting gap near the edge, the effect of the possible topological

edge modes in the normal state. However, as presented in our point-to-point response in the
following, we show that even in these conditions the measurements of the superconducting
edge/corner modes can still provide strong and important evidences for the pairing symmetry in
single-layer FeSe/SrTiO₃. We believe we have addressed all the reviewer's concerns and hope the
reviewer reconsider the publication of our work.

**Comment 1.1:** In the present paper, authors analyze the tunneling conductance of single layer FeSe,
which is in itself a feat. Nevertheless, authors present atomic resolution studies that are only partial,
not reaching the truly interesting edge behavior. Generally, the result is that when approaching a
step authors observe a full gap instead of states within the gap. However, the measurements are not
atomically resolved. Atomic resolution stops quite far from the edges.

**Response 1.1:** We thank the reviewer for raising this point. A clean tip with atomic-level spatial
resolution is very crucial in our experiments because the topologically non-trivial modes, if any,
will be localized around the edge/corner. The atomically resolved image near the edge is indeed
direct evidence supporting the reliability of the measurements.

**Figure R2 | STM topographic image of the (01) edge.** **a**, A copy of Figure 3a in the original
manuscript. **b**, A larger atomically resolved image taken in the area outlined by the blue dashed box
in Figure R2a.

Figure R2a is a reproduction of Figure 3a in the original manuscript, showing an atomically resolved
image about 7.1 nm away from the (01) edge. In Figure R2b, we show a larger atomically resolved
image taken in the area outlined by the blue dashed box in Figure R2a, about 2.1 nm away from the

edge. The atomically resolved image in Figure R2a is part of Figure R2b, and its position is marked
by a red dashed box in Figure R2b. The absence of impurities and defects in Figure R2b suggests
the completeness of the lattice near the edge, as well as the preservation of lattice symmetry.

**In the revised manuscript, the atomically resolved image of Figure 3a has been updated to the image**
**taken in the area outlined by the black solid box in Figure R2b.**

Actually, based on the following facts, it is reasonable to conclude that our spectroscopic
investigations at the edge are atomically resolved:

(1) The atomically resolved images near the edge/corner presented in Figures 2a, 3a and 4b show
that our tips have a spatial resolution at the sub-nanometer scale under current setup conditions (V_s
= 50 mV, $I_t = 500$ pA).

(2) When we collect the spectroscopy map spanning from FeSe to SrTiO₃, the setup conditions for
dI/dV measurements ($V_s = 50$ mV, $I_t = 500$ pA) are the same as those for atomically resolved images
measurements. In addition, the spacing between two adjacent points (0.30 nm for Figure 2c and 0.33
136 nm for Figure 3c) is always less than the lattice constant of single-layer FeSe (0.38 nm).

Therefore, our dI/dV mapping probes the DOS with sub-atomic precision. Despite electronic
disorder or lattice relaxation along edges probably inducing weak contrast, the sub-atomic scale in-
plane resolution is definitely guaranteed.

**Comment 1.2:** Furthermore, the full gap might also be due to changes in the band structure close
to the step. Instead of changes in the in gap conductance, authors observe significant changes in the
gap magnitude. This suggests that the critical temperature is decreasing, which produces very
unfavorable conditions to obtain the bound states proposed in the reference 30.

**Response 1.2:** We agree with the reviewer on the point that the gradual decrease of the
superconducting gap upon approaching the edge implies suppressed T_c at the edge. However, as
aforementioned, the topological property of single-layer FeSe only depends on the pairing signs on
the two electron Fermi pockets centered at the M point, and is unrelated to the superconducting gap
magnitude. Here, for further verification, we demonstrate that the decreased superconducting gap
magnitude will not affect the analysis and conclusion of this work from three aspects:

(1) *Impact of theoretical applicability*—The reduced superconducting gap near the edge can
solely modify the topological edge/corner modes quantitatively and it cannot make them disappear,
as long as the system is in the sign-changed s_{\pm} -wave pairing state. To confirm this, we carry out
numerical simulations based on the model described in the Supplementary Note 5 and we take the
same parameters with those in Figure S8. To simulate the reduced superconducting gap near the
edge, we assume the position-dependent behavior of the superconducting orders in the following
form

$$158 \quad \Delta(r) = \tilde{\Delta}(0.5 + 0.5 * \tanh \frac{r}{R_0}),$$

where r is the distance to the edge, and $\tilde{\Delta}$ is the superconducting order parameter in the bulk ($\tilde{\Delta}$
stands for both the on-site pairing Δ_0 and the next-nearest-neighbor pairing Δ_1). In the above
formula, R_0 characterizes the range near the edge where the superconducting order is suppressed.
One can check that the superconducting gap near the edge is about half of the superconducting gap
in the bulk, which well mimics the experimental results (Figures 2c, 3c, S1d and S1e). We present
the corresponding numerical results in Figure R3. Quantitatively, the Dirac edge modes at the (10)
edge and the Majorana modes at the corner between the (11) and (1 $\bar{1}$) edges maintain. The reduced
superconducting gap near the edge merely modifies the superconducting energy spectrum of the
edge/corner states as shown in Figures R3a and R3b, and makes the corner states more extended as
presented in Figure R3c (which is consistent with the case of larger coherence length near the edge).
We want to emphasize that the reduced superconducting gap near the edge cannot change the
topological property of the system, as long as the superconductor keeps fully gapped. In other words,
if the reduced superconducting gap near the edge changes the topology (the topological property of
the superconductor far away from the edge and the topological property of the superconductor near
the edge are different, such as it being sign-changed s_{\pm} -wave in the bulk and sign-preserving s -wave
near the edge), a topological phase transition must occur and the topological edge/corner modes
must appear at some distance away from the edge correspondingly. However, in the experimental
results presented in the manuscript one can find that, the single-layer FeSe/SrTiO₃ system is fully
gapped everywhere in the real space, indicating the non-existence of such topological phase
transition.

**Figure R3 | Superconducting edge and corner modes in the case where the suppressed**

**superconducting gap near the edges [$\Delta(r) = \tilde{\Delta}(0.5 + 0.5 * \tanh \frac{r}{R_0})$] is considered. **a** shows the**

**superconducting spectrum when open-boundary condition is considered along the (10) direction. **b****

**shows low-energy spectrum when open-boundary condition is considered both along the (11) and**

**($1\bar{1}$) directions. **c** presents the profile of the wave function for the zero-energy modes in **b**.**

(2) *Impact of data analysis*—A direct effect of decrease in T_c at the edge is the increase in the

ratio of the experimental temperature T_{exp} to T_c . When T_{exp}/T_c is close to 1, the zero-bias conductance

(ZBC) of tunneling spectrum will not be zero due to partial condensation of electrons induced by

thermal fluctuations. Figure R4b shows the dependence of the superconducting gap magnitude on

temperature given by the Bardeen-Cooper-Schrieffer (BCS) theory. Then, taking an isotropic gap

function into account, we simulate the temperature-dependent tunneling spectra using the Dynes

formula (Figure R4a). With the increase of T_{exp}/T_c , the coherence peaks become weaker, and ZBC

increases from 0 to 1 (Figure R4c). The increase of ZBC with the increase of T_{exp}/T_c has been

verified by experiments [e.g., Phys. Rev. B **89**, 060506(R) (2014); Nat. Phys. **16**, 536 (2020); Nano

Lett. **22**, 3245 (2022)].

**Figure R4 | Simulated temperature-dependent tunneling spectra.** **a**, Temperature-dependent
 tunneling spectra simulated by the Dynes formula where an isotropic gap function is taken into
 account. **b**, Temperature-dependent superconducting gap magnitude predicted by the BCS theory. **c**,
 Temperature-dependent zero-bias conductance extracted from the tunneling spectrum shown in **a**.

We now discuss the extent to which T_c reduction affects the ZBC in this work: assuming that the
 ratio $2\Delta/k_B T_c$ of single-layer FeSe is a constant, a reduction of nearly half of the superconducting
 gap at the edge (Figures 2c, 3c, S1d and S1e) implies that the T_c at the edge becomes half of 65 K,
 *i.e.*, about 33 K. Here, the critical temperature of 65 K is the gap-close temperature of single-layer
 FeSe, which has been revealed by STM and ARPES investigations [e.g., Nat. Mater. **12**, 634 (2013);
 Phys. Rev. B **89**, 060506(R) (2014); Nat. Commun. **12**, 2840 (2021)]. We can then estimate that the
 T_{exp}/T_c at the edge is about 4.8 K/33 K = 0.15. As indicated by the black arrow in Figure R4c, a
 T_{exp}/T_c of 0.15 does not lead to a significant change in ZBC. In other words, the reduced T_c at the
 edge will not have a significant impact on the zero-bias conductance because the reduced T_c is still
 much larger than the experimental temperature T_{exp} .

The weak effect of increased T_{exp}/T_c on ZBC can also be confirmed by the experimental data of
 temperature-dependent tunneling spectra. Figure R5a is taken from our group's previous work [Phys.

Rev. B **89**, 060506(R) (2014)], showing the temperature-dependent tunneling spectra of single-layer
 FeSe. Figure R5b presents the temperature-dependent ZBC extracted from the spectra shown in
 Figure R5a. Obviously, as T_{exp}/T_c increases, the ZBC first remains unchanged ($T_{\text{exp}}/T_c < 0.25$), then
 gradually increases and approaches 1 ($T_{\text{exp}}/T_c > 0.25$), which is similar to the evolution of ZBC
 shown in Figure R4c. Again, as indicated by the black arrow in Figure R5b, a T_{exp}/T_c of 0.15 does
 not lead to a finite ZBC. Therefore, the decrease of T_c at the edge (from 65 K to 33 K) will not affect
 the detection of edge modes.

 **Figure R5 | Temperature-dependent tunneling spectra of single-layer FeSe. a**, Temperature-
 dependent tunneling spectra collected on single-layer FeSe. The spectra are shifted along the y axis
 at a fixed value of 0.5. **b**, Temperature-dependent zero-bias conductance extracted from the
 tunneling spectrum shown in **a**.

 *Impact of conclusion*—According to the phenomenon that the superconducting gap decreases
 near the edge (Figures 2c, 3c, S1d and S1e), both we and the reviewer agree that T_c at the edge
 decreases, that is, T_{exp}/T_c increases. Based on the analysis above, we have demonstrated that the
 increased T_{exp}/T_c has little contribution to the value of ZBC, that is, the effect of increased T_{exp}/T_c
 on the detection of edge states is very weak. On the other hand, an interesting question is, if our
 estimates are incorrect, what will be the consequences of the finite ZBC induced by the decreased

T_c at the edge? In this case, if one claims to observe a gapless state, then one needs to discuss whether
its origin is a topologically non-trivial mode caused by the sign-changed s_{\pm} -wave pairing, or the
partial condensation of electrons due to large T_{exp}/T_c . *However, this is not our case. Figure 3e*
*clearly shows that zero-bias conductance remains zero at the edge, which unambiguously*
*demonstrates the absence of edge modes* and therefore we do not need to discuss the possible origin
of such non-existent gapless state.

Based on the discussion above, we demonstrate that the decrease of T_c at the edge will not affect the
detection of edge modes from three aspects, and even if it does, it will not affect the analysis and
conclusion of this work—There is no topologically non-trivial edge modes at the edges, which is
solid evidence supporting the sign-preserving s -wave pairing symmetry of single-layer FeSe.

In the revised manuscript, we have added a discussion on the effect of reduced superconducting gap
near the edge on the detection of edge modes before concluding the pairing symmetry of single-
layer FeSe (Lines 161-169). In addition, in the revised Supplementary Information, we have added
Supplementary Note 6 to present the detailed discussion described above.

**Comment 2.1:** Unfortunately, I am not convinced that authors have sufficient evidence to support
their claim of absence of s_{\pm} superconductivity in this system. Furthermore, writing of the paper is
such that it seems that authors claim that s_{\pm} superconductivity is absent in many other iron based
systems too, which is quite unjustified.

**Response 2.1:** The reviewer may have misunderstood our view of the pairing symmetry of iron-
based superconductors. Our work only confirms that the pairing symmetry of single-layer FeSe is
the sign-preserving s -wave, under the unique Fermi surface consisting only of electron pockets
located at the M point. For other iron-based superconductors with Fermi pocket at the Γ point
participating in pairing, the theory is inapplicable. It is important to note that we do not draw
conclusions and speculate on the pairing symmetry of other iron-based superconductors.

In the first paragraph of the manuscript, we give a fair review of the controversy over the pairing
symmetry of heavily electron-doped iron chalcogenides in recent years, and list the leading ideas,
namely, sign-changed s_{\pm} -wave and sign-preserving s -wave. In addition, quite a few of the references

we cited support or favor the sign-changed pairing symmetry, such as, Nat. Phys. **14**, 134 (2018),
Phys. Rev. B **98**, 134503 (2018), Phys. Rev. Lett. **123**, 036801 (2019), Nano Lett. **19**, 3464 (2019),
Commun. Phys. **3**, 75 (2020). At the end of the paragraph, we remain neutral and believe that the
pairing mechanism of these materials is still an open issue.

In the following paragraph, we investigate the pairing symmetry of single-layer FeSe from a
topological point of view, which is completely different from the experimental principles of
previous work and thus promising to settle the long-lasting debate. Taking the unique lattice
structure and spin-orbit coupling into account, a sign-changed s_{\pm} -wave pairing leads to a second-
order topological state while the sign-preserving s -wave pairing results in a topologically trivial
state. Here, our spectroscopic investigations at isolated edges and corners demonstrate the non-
existence of edge/corner modes, which provides strong evidence for the sign-preserving s -wave
pairing in single-layer FeSe. Note that in all the discussion, we do not generalize the analysis of the
experimental results and the conclusions about the pairing symmetry to other iron-based
superconductors.

In the last paragraph, we prospect that the method could be used to study the pairing symmetry of
other heavy heavily electron-doped iron chalcogenides. Our outlook does not evaluate experimental
results or predict possible conclusions.

In summary, we are keeping an open mind about the pairing symmetries of other iron-based
superconductors.

In the revised manuscript, we frankly point out that the pairing symmetry of the iron-based
superconductor with space group $P4/nmm$ is sensitive to the electron pockets centered at the M point
(Lines 62-64). In addition, we emphasize the applicability of this technique to the study of pairing
symmetries in heavily electron-doped iron chalcogenides (Lines 61-62). Finally, we make it clear
that we only discuss the pairing symmetry of single-layer FeSe from the topological perspective in
this work (Lines 77-78).

**Comment 2.2:** Unless a very large additional set of data is provided, showing clearly what happens
to the band structure in this system in presence of edges, I do not think that this paper can be
published.

**Response 2.2:** We thank the reviewer for pointing out the issue that, whether the normal-state edge
 modes can affect the topological superconductivity in the sign-changed s_{\pm} -wave pairing state. In
 fact, *the normal-state bands in the presence of edges cannot change the correspondence between*
 *the topological superconductivity and the pairing symmetry*. One can understand this from the
 following two aspects.

**Figure R6 | Energy band of single-layer FeSe.** **a**, the bands for single-layer FeSe in the absence
 of the spin-orbit coupling. In the calculation, we use the model Hamiltonian (all five d orbitals of
 the Fe atom are taken into account) and parameters in Appendix G of Ref. [30]. The blue-dashed
 line labels the Fermi energy. **b**, the bands corresponding to the case in **a** with open boundary
 condition along the (10) direction. In calculating **b**, the spin-orbit coupling between the five d
 orbitals, *i.e.*, the atomic spin-orbit coupling $\lambda \mathbf{L} \cdot \mathbf{S}$ with \mathbf{L} being the orbital angular momentum and
 \mathbf{S} the spin angular momentum, are considered and the strength of the spin-orbit coupling is taken to
 be $\lambda = 80 \text{ meV}$. By considering the sign-changed s_{\pm} -wave pairing in the bands in **b**, one obtains the
 superconducting Dirac edge modes on the (10) edge shown in Figure 10 in Ref [30].

 Firstly, in the theoretical work in Ref. [30], the effect of the normal-state edge bands has been taken
 into account in simulating the topological superconductivity in the sign-changed s_{\pm} -wave pairing
 state based on a genuine model for single-layer FeSe with all five d orbitals of the Fe atom
 considered. Here, in simulating the topological superconductivity, the following BdG Hamiltonian
 is used

$$314 \quad H_{BdG} = \begin{pmatrix} H_0(k) & H_{sc}(k) \\ H_{sc}^\dagger(k) & -H_0(k) \end{pmatrix}$$

where we take the same basis with that in Ref. [30]. In the above, $H_0(k)$ is the normal-state
 Hamiltonian for single-layer FeSe in the presence of the spin-orbit coupling among the five d

orbitals, and $H_{sc}(k)$ is the superconducting pairing. In calculating the topological superconducting
edge modes, we consider the open-boundary conditions in both $H_0(k)$ and $H_{sc}(k)$; And the normal-
state edge bands have been faithfully reflected in $H_0(k)$. In Figure R6, we show the normal-state
bands in the periodic and open-boundary conditions calculated based on $H_0(k)$ for single-layer
FeSe; by taking $H_{sc}(k)$ into account in the open-boundary bands shown in Figure R6b, one obtains
the superconducting Dirac edge modes on the (10) edge, which is shown in Figure 10 in Ref. [30].
Please note that the only difference between the calculations here and those in Ref. [30] is the spin-
orbit coupling strength (80 meV for here and 40 meV in Ref. [30]), which, however, does not change
the conclusion. Here, we also want to point out that for iron chalcogenides, the first-principle
calculations cannot give the right band structures and we tune the parameters in $H_0(k)$ by hand to
fit the bands observed in the ARPES measurements.

Secondly, if single-layer FeSe/SrTiO₃ has different topological properties between the bulk and the
edge, which may be due to the modification of the normal bands or the superconducting order near
the edge, one cannot get full-gap superconducting energy spectrum everywhere in the real space
because a topological phase transition must occur and the topological edge/corner modes must
appear at some distance away from the edge (Please also see **Response 1.2**). However,
experimentally we find that the single-layer FeSe/SrTiO₃ system is fully gapped everywhere in the
real space, meaning that no such topological phase transition occurs.

Finally, according to the **Comments 3.5** and **3.6**, we notice that the reviewer thinks there may be
magnetism near the edge. As shown in **Response 3.6**, in general the magnetism can indeed break
the topological superconductivity verified in Ref. [30] for the possible symmetry-breaking effect
and affect our conclusion on the pairing symmetry of single-layer FeSe/SrTiO₃. However, currently
there is no smoking-gun experimental evidences for the coexistence of superconductivity and
magnetic order (no matter in the bulk or near the edge) in single-layer FeSe/SrTiO₃.

**Comment 2.3:** In re-writing, authors might consider all possible explanations for their results and
state more clearly what is understood and favored by their data and what should be studied on later
work.

**Response 2.3:** We thank the reviewer for this kind comment. In the revised version, the mechanism
and application scope of this technique are more clearly indicated (**Lines 61-64, 77-78**). Furthermore,

the effects of actual factors beyond the theoretical model on the detection of edge/corner modes,
such as the size of edge/corner (Lines 107-111, 132-135, Supplementary Note 2), the decrease in
superconducting gap near the edge (Lines 161-169, Supplementary Note 6) and the energy
resolution of STS measurements (Lines 172-173, Supplementary Note 7), which are not considered
in the previous version, are discussed in detail now. Finally, we attach some descriptions of the
effects of magnetism on determining topological properties of single-layer FeSe (Lines 218-221),
which is worthy of further investigation in later work.

**Comment 2.4:** In addition, I do not feel that the data present the expected degree of novelty.

**Response 2.4:** Topology has been a hot topic in recent years. The coexistence of topologically non-
trivial states and superconductivity has been found in more and more materials. These novel
phenomena are indeed very eye-catching and contribute to the exploration of topological
superconductivity.

In this work, our spectroscopic investigations demonstrate that the superconducting states at isolated
edges and corners of single-layer FeSe are topologically trivial. At first glance, our results are not
impressive. *However, it is important to emphasize that, our results have rich physical implications.*
According to our solid theoretical calculations [Phys. Rev. X **12**, 011030 (2022)], a sign-changed
s_{\pm} -wave pairing leads to a second-order superconducting state at the (01) edge and at the corner
between the (11) and $(1\bar{1})$ edges, while sign-preserving s -wave state remains topologically trivial
even in the presence of inversion symmetric Rashba SOC. Based on the correspondence between
topological properties and pairing symmetries, our work provides the first definite experimental
evidence for the sign-preserving s -wave state, which settles the long-lasting debate between the s -
and s_{\pm} -wave pairing, and is an important step towards revealing the microscopic pairing mechanism
of iron chalcogenides.

**Comment 3.1:** To aid authors in re-writing and submit somewhere else to a more specialized venue,
I would suggest to consider the following aspects:

**Response 3.1:** We thank the reviewer for these valuable comments. These suggestions have
improved the quality of our work. We have revised our manuscript according to the comments above
and the specific suggestions below.

We need to emphasize that our analysis is solid and that our findings are important. *We investigate*
*the pairing symmetry of single-layer FeSe from a topological point of view, which is completely*
*different from the experimental principles of previous work* and thus promising to settle the long-
lasting debate between the s - and s_{\pm} -wave pairing. Our work will attract a lot of attention and follow-
up studies in the superconductor community, and is therefore suitable for publication in Nat.
Commun.

**Comment 3.2:** - Authors should specify clearly the applicability of the approach of ref 30. While
this is in principle quite generic, here it is applied to single layer FeSe and it is not clear to me how
this would influence other iron based materials. Authors should be much more careful in specifying
that their results refer to FeSe single layers.

**Response 3.2:** We thank the reviewer for this important comment. As aforementioned, the
topological property of the system only depends on the lattice structure (the P4/nmm space group),
the electronic structure (the Fermi pockets on the BZ boundary), and the pairing structure (the
pairing signs on the two Fermi pockets at the M point). For the iron-based superconductor, the lattice
structure (we specify it to the P4/nmm symmetric system) and the electronic structure are fixed.
Therefore, the topological property is solely determined by the pairing signs on the two Fermi
pockets at the M point. Please note that the topological superconducting state in Ref. [30] has
nothing to do with the with the γ Fermi surface (Figure R1a). *That is, the topological*
*superconducting state can detect the pairing signs on the two electron Fermi pockets at the M*
*point, i.e., the pairing phase on the α and β Fermi surfaces (Figure R1) in both the iron*
*chalcogenides and the iron pnictides, but cannot provide information on the pairing phase on the*
*γ Fermi pockets at the Γ point.*

As for heavily electron-doped iron chalcogenides such as $A_x\text{Fe}_2\text{Se}_2$ (A = alkali metal), $(\text{Li}_{1-x}\text{Fe}_x)\text{OHFeSe}$
and single-layer FeSe on SrTiO_3 , the Fermi surface consists only of two electron
pockets centered at the M point. Therefore, we can apply this newly developed technique to
determine the pairing symmetry of these iron chalcogenides. Among these electron-doped iron-
based superconductors, single-layer FeSe is the most ideal material for searching the edge/corner
modes, because the high-quality edges can be easily constructed.

In the revised manuscript, we have pointed out that the topological properties of an iron-based
superconductor with a centrosymmetric space group $P4/nmm$ are only sensitive to the pairing signs
of two electron pockets centered at the M point (Lines 62-64). In addition, we list several iron-based
superconductors that may hold topologically non-trivial edge/corner states, which are worthy of
further investigation (Lines 215-217).

**Comment 3.3:** - Authors should better explain what a “second order superconducting state” means.
And, in particular, why such a “second order state” would be better than the effects that are being
studied using the “first order state”. The argument that impurities produce scattering bound states
with an inherent complexity and that instead symmetry based arguments could help out is very
difficult to hold. One needs to create a bound state also when considering symmetry arguments
(actually, the usual bound states can be traced down to symmetry arguments too). The interaction
of the edge with the rest of the system does play a role in the creation of a bound state. To me, there
is no fundamental difference when it comes to creating in-gap modes.

**Response 3.3:** We thank the reviewer for the comment and suggestion.

Firstly, the “second-order superconducting state” in the manuscript actually refers to the “second-
order topological superconducting state” (to avoid any ambiguity we have changed the expression
“second-order superconducting state” to “second-order topological superconducting state” in the
revised manuscript). The second-order topological superconducting state is a well-studied
topological state in recent years, which is inspired by the study of the second-order topological
insulators. Generally, we call a d -dimensional (dD) superconductor a second-order topological
superconductor, if it is fully gapped in the bulk and supports $(d-2)D$ zero-energy or gapless modes,
such as the 2D superconductor supporting the zero-energy corner modes (like the state in Ref. [30])
and the 3D superconductor supporting the gapless helical or chiral hinge modes (a hinge means the
crossing line between two surfaces). The mass domain picture shown in Ref. [30] is crucial in
realizing the second-order topological superconducting state. Currently, there are mainly two ways
to get a second-order topological superconductor. One way is based on the edge modes of the
topological insulators. Specifically, by introducing superconductivity and/or magnetic orders
through the proximity effect, one can realize the mass domain picture in Ref. [30] on the edge modes
of a topological insulator. Such kind of second-order topological superconductors are the extrinsic

ones, where usually only the $(d-2)$ D zero-energy or gapless modes are supported. Many theoretical
 proposals have been proposed for the extrinsic second-order topological superconductors, such as
 introducing the d -wave superconductivity into a 2D topological insulator, or introducing the BCS-
 type superconductivity and stripe magnetic order into a topological insulator. The other way to
 realize a second-order topological superconductor is to find superconductors with special band
 structures and special pairing structures. Such kind of second-order topological insulators are the
 intrinsic ones and are protected by the crystalline symmetries; Moreover, in the intrinsic second-
 order topological superconductors there usually exist both the topologically nontrivial $(d-1)$ D edge
 modes and the $(d-2)$ D corner modes. The state in Ref. [30] is a typical intrinsic second-order
 topological superconducting state.

**Figure R7 | In the strong scattering potential condition, the impurity induces a small region**
 **which is topologically trivial.** Namely, the impurity creates a small finite-size inner circular edge
 (right figure). The edge in the left figure corresponds to a circular with infinite large radius.

Secondly, we want to point out that we never claim, “such a second order state would be better than
 the effects that are being studied using the first order state” nor the method used in our work is better
 than others. To trace out the pairing symmetry of single-layer FeSe/SrTiO₃, many experimental
 studies have been carried out to probe the impurity bound states, especially the bound states induced
 by the nonmagnetic impurity. According to Anderson’s theorem, the nonmagnetic impurity cannot
 induce in-gap bound states if a superconductor has sign-preserving pairing symmetry. However, the
 current results on the impurity bound states in the single-layer FeSe/SrTiO₃ are still elusive and no
 consensus has been reached [e.g., Nat. Phys. **11**, 946 (2015); Phys. Rev. B **97**, 024502 (2018); Phys.
 Rev. Lett. **124**, 097001 (2020) favor sign-preserving s -wave state in single-layer FeSe, while Phys.
 Rev. Lett. **123**, 036801 (2019); Nano Lett. **19**, 3464 (2019); Commun. Phys. **3**, 75 (2020); Nano
 Lett. **22**, 3245 (2022) support sign-changed pairing symmetry in single-layer FeSe]. Such
 controversy arises from the facts that, experimentally it is difficult to determine the type (magnetic

or nonmagnetic) and the scattering strength of the impurity, both of which are vital to come to the
correct conclusion. In addition, the location of the impurity in a superconductor can also affect the
results [e.g., Phys. Rev. B **94**, 134502 (2016); Nat. Phys. **14**, 134 (2018)]. Regarding to the role of
the impurity and the edge in inducing the in-gap modes, we agree with the reviewer's arguments to
some extent. In our opinion, in the strong scattering potential condition the impurity creates a small
circular edge in the system as sketched in Figure R7. In this case, the topological edge modes still
exist but they are discretized into a series of energy levels, and the lowest energy level has energy
proportional to $1/r$ with r the radius of the inner circular edge in Figure R7. Nevertheless, our study
does have some unique points——*If the single-layer FeSe/SrTiO₃ is in the sign-changed s_{\pm} -wave*
*pairing state, the experimental measurements at different edges, i.e., the (01) and (11) edges,*
*and the measurements at the corner between the (11) and (1 $\bar{1}$) edges, should all be consistent*
*with each other.* This imposes rather strong constraints when we conclude the pairing symmetry
according to the experimental observations.

**Comment 3.4:** - Authors state that “obviously, Delta's near the (11) edge are smaller than those in
the bulk, which may be due to lattice discontinuity. Nevertheless, the zero-bias conductance, our
main concern, is uniform in real space.” I completely disagree with such an approach. The zero bias
conductance is very much connected to the rest of the tunneling conductance in the superconducting
phase. Changing T_c does lead to changes that can easily preclude the appearance of in-gap modes.
Along this line, the change in T_c is not considered when authors discuss the role of the substrate in
the detection of edge states.

**Response 3.4:** We thank the reviewer for raising this point. This comment mainly concerns the
effects of the change in superconducting gap/ T_c at the edge on the detection of the edge modes,
which is similar to the **Comment 1.2**. In **Response 1.2**, we have demonstrated from both theoretical
and experimental points of view that the decrease of superconducting gap/ T_c at the edge does not
affect our conclusion that the edge mode does not exist. Please refer to **Response 1.2** for our detailed
discussion of the effects of decreased T_c at the edge on the detection of edge modes. In addition, we
have shown that the normal-state bands in the presence of edges cannot change the correspondence
between the topological superconductivity and the pairing symmetry (please refer to **Response 2.2**
for details).

In the revised manuscript, we have added a discussion on the effect of reduced superconducting gap
near the edge on the detection of edge modes before concluding the pairing symmetry of single-
layer FeSe (Lines 161-169). In addition, Supplementary Note 6 has been added to the revised
Supplementary Information to provide a detailed discussion from the perspectives of numerical
simulations and temperature-dependent STS experiments.

**Comment 3.5:** - Authors present “anomalies induced by local defects.” But what is the origin of
the in-gap modes created by such local defects if it is not some sort of gap-varying behavior? There
is one defect where the gap is completely suppressed (left white box of Fig 2c).

**Response 3.5:** Anomalies in the superconducting gap and zero-bias conductance are observed in
the same region marked by the white boxes in Figures 2c and 2d, which may be caused by local
defects (white box in Figure 2a). Figure R8a is a reproduction of Figure 2a in the original manuscript.
The atomically resolved image presented in Figure R8a shows bright lobes on adjacent top-layer Se
sites, which may be caused by some perturbation at the Fe site [Phys. Rev. B **90**, 134520 (2014);
Nano Lett. **16**, 4224 (2016)]. Figure R8b shows the tunneling spectrum collected at the green cross
near the local defect. The tunneling spectrum has a conductance peak at -4.0 mV (dashed line),
which is similar to the characteristics of Se_{Fe} antisite defect [Phys. Rev. B **97**, 024502 (2018)].
Therefore, we preliminarily attributed the anomalies on the topographic image to Se_{Fe} antisite defect.

**Figure R8 | In-gap bound states induced by local defects.** a, A copy of Figure 2a in the original
manuscript. b, The tunneling spectrum collected at the green cross shown in a.

The successful capture of anomalies in topographic image and electronic state proves that our
equipment (or setup conditions) has sufficient spatial resolution and energy resolution to detect
potential edge/corner modes. On this basis, no obvious anomalies in topographic image and
superconducting gap are observed near the (01) edge (Figures 3a and 3c), indicating the high quality
of the edge and providing an excellent opportunity to study the edge modes. Therefore, the position-
independent zero-bias conductance (Figures 3d and 3e) intrinsically reveals the absence of
topologically non-trivial edge modes and provides solid evidence for the sign-preserving s -wave
pairing symmetry of single-layer FeSe.

As discussed above, *the exact origin of the local defects is not the focus of this work and is*
*irrelevant to the topological properties studied here.* How these defects interact with electrons to
generate bound states is beyond the scope of this paper. Actually, the mechanism of the interaction
between defects and electrons to produce bound states is complex and requires further specific
studies. We would like to emphasize that the experimental results of impurity-scattering
investigations performed by various groups on single-layer FeSe are irreconcilable to date [e.g., Nat.
Phys. **11**, 946 (2015); Phys. Rev. B **97**, 024502 (2018); Phys. Rev. Lett. **123**, 036801 (2019);
Commun. Phys. **3**, 75 (2020)]. *It is because the powerful technique of impurity scattering has*
*encountered some controversy that we have developed this new method from a topological point*
*of view to settle the debate between s - and s_{\pm} -wave pairing.*

In the revised manuscript, we have modified the discussion part of the main text (Lines 169-173)
and added a Supplementary Note 7 to the revised Supplementary Information, which preliminarily
discuss the possible origin of defects and demonstrate the ability of our equipment to capture
edge/corner modes.

**Comment 3.6:** - Along the line of thought of previous point, authors might argue that these local
defects are magnetic. Is then magnetism completely absent from edges? This is very difficult to
understand.

**Response 3.6:** As explained in the **Response 3.5**, the mechanism by which these local defects
produce bound states is beyond the scope of this paper. Whether these defects are magnetic or not,
they will not affect our study of the topological properties at the edge/corner. Therefore, we do not

discuss the magnetism of these defects. In our opinion, it is difficult to exactly determine the
magnetic properties of defects with the data currently available. Sometimes a nominally
nonmagnetic impurity may pin local magnetism of the parent material [Phys. Rev. Lett. **86**, 296
(2001)], and sometimes a commonly perceived magnetic element appears to lose its local moment
when embedded in the parent material [Phys. Rev. B **93**, 014507 (2016)].

As to the possible magnetism on the edge, it is an interesting issue and we understand the reviewer's
concern. First, we must admit that in the current work we cannot tell whether there is magnetism or
not near the edge, and to answer this question, spin-polarized STM measurements should be
performed, which is beyond the capability of our equipment. However, we want to point out that
even if there is magnetism on the edge, it is not likely to come from the possible magnetic impurity
pointed out by the reviewer in the **Comment 3.5**. Usually, the impurity affects the superconductivity
locally [e.g., Phys. Rev. B **97**, 024502 (2018)]. This can also be verified by the fact that, the
impurities merely suppress the superconducting gap and lead to finite zero-bias conductance in a
rather local region as presented in Figures 2c and 2d in the main text respectively. The
superconducting gap retains its bulk value quickly as long as it is off the impurities. As shown in
Figure 2a, the impurity closest to the (11) edge is about 7.5 nm away from the edge. Therefore, the
impurity is unlikely to induce magnetism at the (11) edge and has little effect on the electronic state
at the edge. As for the (01) edge which is expected to support the Dirac edge modes in the sign-
changed s_{\pm} -wave pairing state, there is no impurity nearby as shown in Figure 3a in the manuscript.
Given the above facts, it is hard to imagine that the possible magnetic impurities indicated by the
reviewer could induce magnetism at the edge.

If there is magnetism at the edge, it is more likely to come from other mechanisms such as the
possible long-range magnetic order as indicated by the reviewer in **Comment 3.10** (Ref. [15] in the
revised Supplementary Information). Actually, the effect of the long-range magnetic order on the
topological superconductivity is complicated. The type of the magnetic order (such as ferromagnetic,
antiferromagnetic, etc.), and the strength and direction of the magnetic polarization, could all have
non-neglectable effects. Generally, the symmetries of lattice required by the topological
superconductivity can hardly preserve in the presence of the magnetic order, and the topological
superconductivity in the theoretical work in Ref. [30] can hardly maintain correspondingly.
Consequently, the possible long-range magnetic order can indeed affect our conclusion on the

pairing symmetry of the single-layer FeSe/SrTiO₃. However, it is worth pointing out that currently
there is no certain signatures for the coexistence of superconductivity and magnetic order in single-
layer FeSe. We also want to remind that *in Ref. [15] in the revised Supplementary Information*
*mentioned by the reviewer in the last comment, the authors did not find any direct evidence for*
*the long-rang magnetic order*. Actually, the experiment only detects evidences for the edge state
that lies ~ 100 meV below the Fermi surface, and the authors *assume* a checkerboard
antiferromagnetic order to simulate the bands observed in the ARPES measurements (Please notice
that this is not the direct evidence for the magnetic order).

In the revised manuscript, we attach some descriptions of the effects of magnetism on determining
topological properties of single-layer FeSe (Lines 218-221), which is worthy of further investigation
in later work.

**Comment 3.7:** - In Figs 2e and 3e, authors should plot the normalized tunneling conductance, not
the absolute value. This would allow the reader to understand how close that is to “zero”.

**Response 3.7:** We thank the reviewer for this constructive comment. In the revised manuscript, we
plot both the raw and the normalized tunneling conductance to show the uniformity of density of
states and how close the density of states is to “zero”. The normalization method has been described
in Supplementary Note 1. As shown in Figure R9, both the raw and the normalized zero-bias
conductance remain almost constant with moving close to the isolated (11)/(01) edges. The zero-
bias conductance difference in Figures R9a and R9b could be due to varied tunneling contribution
from the substrate (related to the coverage and distribution of FeSe films) and tip apexes.

**Figure R9 | Column-averaged zero-bias conductance as a function of the distance to the (11)**
**edge (a) and to the (01) edge (b).**

In the revised manuscript, Figures 2e and 3e have been updated accordingly.

**Comment 3.8:** - It is quite disturbing and time consuming to see that authors chose red to represent
the tunneling conductance at the edge, and the same color for the size of the gap far from the edge.

**Response 3.8:** We thank the reviewer for raising this point. We have changed the color as suggested
by the reviewer.

In the revised Figures 2 and 3, we choose red to represent the tunneling conductance far from the
edge, and the same color for the size of the gap far from the edge.

**Comment 3.9:** - Authors use a finite Dynes term, which is just between 100 and 200 microeV large.
This is less than one per cent of the gap magnitude. It can probably be removed without changing
too much the rest of the parameters.

**Response 3.9:** We thank the reviewer for this pertinent comment. We re-fit the tunneling spectra
using the Dynes formula without the broadening factor Γ . As expected, the rest of the parameters do
not change much after the broadening factor was removed. Figure R10 shows the new fitting results,
from which it can be found that gap magnitude decreases with moving close to the edge.

**Figure R10 | Fitting of tunneling spectra.** a, A copy of the blue spectrum at the bottom of Figure
3b in the main text, and its background (red dashed curve). b, Normalized data (blue circles) and
the fitting result (red curve). c, Fitting results of normalized tunneling spectra with various

superconducting gap magnitude. **d-e**, Averaged gap magnitude as a function of the distance to the
(11) and (01) edges.

In the revised Supplementary Information, we have updated the Figure S1 and Table S1, and
modified the Dynes formula. We have also deleted the description of the broadening factor Γ . In
addition, we have re-simulated the tunneling spectra in Figure S14f using the Dynes formula without
the broadening factor Γ , and then re-generated the normalized tunneling spectrum in Figure S14g.

**Comment 3.10:** - I also disagree with the discussion about Reference 7. There, it is shown that one
has a topological edge state and superconductivity. This shows that there are relevant band structure
changes at the edge and that these cannot be neglected and should be incorporated in the discussion
about what happens at the edge. What is the influence of such changes in the “second order
superconducting state”? Clearly, there is an influence, because T_c changes and I do not understand
why in such conditions one should expect a “second order” effect to hold.

**Response 3.10:** We thank the reviewer for this critical comment. Yes, the previous work [Nat. Mater.
15, 968 (2016)] has investigated the topological properties of single-layer FeSe and discovered the
1D topological edge states at the M point of the Brillouin zone. However, we want to emphasize
again that the topological phase in Ref. [Nat. Mater. 15, 968 (2016)] is associated with the spin-
orbit coupling gap in the antiferromagnetic state, and the topological band lies ~ 100 meV below
the Fermi surface, which is well beyond the energy window of the superconducting gap (~ 15 meV).
Therefore, the topological edge states revealed in Ref. [Nat. Mater. 15, 968 (2016)] are not related
to the specific properties of superconductivity and cannot reflect the relevant information of
superconducting pairing.

We agree with reviewer that Ref. [Nat. Mater. 15, 968 (2016)] indicates that there may be relevant
band structure changes at the edge. Therefore, according to the reviewer’s suggestion, we
investigate the band structure of FeSe in presence of edges (please refer to **Response 2.2** for details).
It is found that the normal-state bands in the presence of edges cannot change the correspondence
between the topological superconductivity and the pairing symmetry. As for the magnetism at the
edge, we want to emphasize that there is no conclusive evidence that single-layer FeSe has long-
range magnetic order (please also refer to **Response 3.6**).

We also understand the reviewer's concern about the reduction of the superconducting gap at the
edge. Therefore, we carry out numerical simulations based on the model described in the
Supplementary Note 5, where a superconducting order $\Delta(r) = \tilde{\Delta}(0.5 + 0.5 * \tanh \frac{r}{R_0})$ is taken to
mimic the experimental results (please refer to **Response 1.2** for details). As shown in Figure R3,
the reduced superconducting gap near the edge cannot change the topological property of the system.
On the other hand, the decrease of gap magnitude near the edge/corner implies the reduction in T_c ,
which may affect the spectral weight and make edge/corner modes difficult to identify. It is
estimated that T_c at the edge drops by about half if we assume that the ratio $2\Delta/k_B T_c$ of single-layer
FeSe is a constant. In this case, due to the reduced T_c of 33 K is still much larger than the
experimental temperature of 4.8 K, no anomaly is expected on the zero-bias conductance (please
refer to **Response 1.2** for details).

Actually, the second-order topological superconducting state depends on the lattice structure, the
electronic structure and the pairing structure of the system. In the case of iron-based superconductors
with space group P4/nmm, such topological superconducting state is only related to the pairing signs
on the electron Fermi pockets at the M point. As the third reviewer said, “as soon as there is a sign
change in the *s*-wave SC order parameter components on the two Fermi surfaces at the M point, the
state is inevitably topologically non-trivial—regardless any other system parameters”. We
emphasize that these finding is not only verified by a minimal lattice model, but also has been
confirmed by a more realistic model with all the five *d* orbitals from Fe taken into account
(Appendix G in Ref. [30]).

In the revised version, the effects of factors mentioned above on the detection of edge/corner modes,
such as the magnetism of edge (Lines 218-221), decrease in gap magnitude and reduction of T_c at
edge (Lines 161-169, Supplementary Note 6), are added.

-----

Response to the Reviewer #2

-----

In the manuscript by Z. Wei et al. submitted to Nature Communications, the authors investigate the
superconducting pairing symmetry in monolayer FeSe grown on STO using scanning tunneling
spectroscopy (STS) measurements. Many groups have investigated the enhanced superconducting
properties of monolayer FeSe on STO by various techniques but conclusive evidence for the pairing
symmetry has been elusive. Being able to resolve this issue will help further our understanding of
the enhanced superconducting state as well as superconductivity in the iron pnictides in general and
hence suitable for Nature Communications. Taking advantage of the FeSe structure and the location
of the inversion center in the monolayer, the authors show how different pairing symmetries give
rise, or not, to topologically protected spectral features (i.e. Majorana modes) at specific domain
edges and corners. While confirmation of zero bias peaks being Majorana modes is a tricky task for
topological superconductors, here the authors simply look for the existence of a zero bias peak and
the lack of one can be connected to a specific s-wave pairing symmetry. Reproducibility is key for
such an investigation and the authors have revealed their results are consistent across different edges
and corners on the sample. Hence the authors use topology to discriminate between the different
pairing symmetries with high confidence. The approach is sound, and the results are indeed
interesting. I recommend the manuscript for publication in Nature Communications in its current
form.

**Authors:** We thank the reviewer for his/her positive comment, recognition and recommendation.

The reviewer has fully caught the breakthrough and innovation of this work.

Determining the pairing symmetry of single-layer FeSe on SrTiO₃ is the key to understanding the
enhanced pairing mechanism and establishing the exact microscopic pairing model. Despite
significant efforts including extensive impurity-scattering investigations and quasiparticle
interference measurements, it remains controversial whether the pairing symmetry of single-layer
FeSe is the sign-preserving s- or the sign-changed s_±-wave. Therefore, we develop a promising way
from a topological point of view to settle this long-lasting debate. Based on the unique lattice
structure and the Rashba-type spin-orbit coupling, we found that a sign-changed s_±-wave pairing

leads to a second-order superconducting state which supports gapless Dirac modes at the (01) edge
and Majorana zero-energy modes at the corner between the (11) and ($\bar{1}\bar{1}$) edges, while the sign-
preserving *s*-wave states remain topologically trivial. Inspired by these theoretical results, we
fabricate high-quality sub-monolayer FeSe by molecular beam epitaxy, and then characterize the
superconducting states at isolated edges and corners by low-temperature scanning tunneling
microscopy/spectroscopy.

In order to avoid misunderstandings such as those in **Comment 3.3** raised by the first reviewer, it
is important to clarify that we have never claimed that the method used in this work is better than
the impurity scattering method. Impurity scattering method is a powerful tool for studying the
pairing symmetry and is applicable to almost all superconductors. However, in terms of the pairing
symmetry of single-layer FeSe, the results obtained by impurity-scattering investigations are
different among different groups. The reasons for the controversy are complex. One possibility is
that the magnetism of the impurities is difficult to be determined. Therefore, we carry out this work
from a topological point of view based on a different principle. Our method also has some
limitations, such as being only applicable to iron-based superconductors and only being able to
determine the phase difference of the Fermi pockets centered at the M point of the Brillouin zone.
And this is why we have placed limitations on superconducting materials when extending this
experimental method (last paragraph of the main text). In addition, with regard to similar
misunderstandings in **Comment 2.1** raised by the first reviewer, we emphasize that at this stage we
do not give any comment or conjecture on the pairing symmetry of these iron-based superconductors.
According to the constructive comments given by the first and the third reviewers, we have enriched
the content of our manuscript, including the discussions on the ability to capture edge/corner modes,
the effects of reduced T_c at the edge, the validity of theoretical predictions, etc.

The revised manuscript retains all the key points and major findings. We believe that the reviewer
will be satisfied with the revised manuscript and continue to recommend the manuscript for
publication in Nature Communications.

-----

Response to the Reviewer #3

-----

The authors use an STM technique on single-layer FeSe on a SrTiO₃ substrate with the eventual
goal to discern between two superconducting pairing candidates: an s_{\pm} -wave or an s_{++} -wave state
where the pairing sign on the two Fermi surfaces is changed or preserved, respectively. The
particular intent is based on the work by parts of the authors in Ref. 30 [S. Qin et al., Phys. Rev. X
12, 011030 (2022)], where they predicted the s_{\pm} -wave state to be a second-order topological SC
state with Dirac cones and Majorana states on particular edges and corners—while the sign-
preserving s_{++} -wave state is non-topological. Using STM the authors study various edges and
corners of their sample. They do not observe any signatures of the Dirac cones or Majorana states,
leading to their conclusion that the pairing state should be the non-topological sign-preserving s_{++} -
wave state.

The paper is written in a comprehensible way, and the results are clearly presented and
communicated. Furthermore, the results contribute to the understanding of the pairing symmetry in
this highly-debated iron-based SC, and they are of interest for a broader audience. If the authors can
convincingly answer my questions, I would recommend a publication in Nature Communications.

**Authors:** We appreciate the reviewer's detailed review of our work. The reviewer has fully caught
the key point of our manuscript, which is to determine the pairing symmetry of single-layer
FeSe/SrTiO₃ from a topological point of view.

We are grateful to the review for his/her very positive comments and recommendation of this work.
Here, we innovatively determine the sign-preserving s -wave state of single-layer FeSe based on its
topological properties, which provides a paradigm for studying the pairing symmetry of other
heavily electron-doped iron chalcogenides. Therefore, our work is an important step towards fully
revealing the microscopic pairing mechanism of iron chalcogenides.

In the following, we present a detailed response to the questions raised by the reviewer. We believe
we have solved all the reviewer's questions and concerns.

**Comment 1:** From a view into Ref. 30 and Eq. (8) therein, I understand that as soon as there is a
sign change in the s -wave SC order parameter components on the two Fermi surfaces at the M point,
the state is inevitably topologically non-trivial—regardless any other system parameters—, is that
correct? (If so, one could maybe emphasize that in the paper.)

**Response 1:** We thank the reviewer for raising this point. Yes, it is exactly as the reviewer said, “as
soon as there is a sign change in the s -wave SC order parameter components on the two Fermi
surfaces at the M point, the state is inevitably topologically non-trivial—regardless any other system
parameters”. Eq. (8) in Ref. [30] gives a Fermi surface criterion for the topological superconducting
state in the weak pairing limit. As shown in our response to the comment raised by the first reviewer
(please refer to **Lines 28 – 95** in this document for details), the theory in Ref. [30] is actually solid
and general. In short, the topological property of the system only depends on the lattice structure
(the symmetry space), the electronic structure (the Fermi pockets on the BZ boundary), and the
pairing structure (the pairing signs on the Fermi surfaces). For the iron-based superconductor, the
lattice structure (we specify it to the $P4/nmm$ symmetric system) and the electronic structure are
fixed. Therefore, the topological property of single-layer FeSe/SrTiO₃ is solely determined by the
pairing signs on the two Fermi pockets at the M point.

In the revised manuscript (Lines 62-64), we have emphasized the applicability of our technique as
follow:

“Our theoretical work³⁰ shows that the topological properties of an iron-based superconductor with
a centrosymmetric space group $P4/nmm$ are only sensitive to the pairing signs of the two electron
pockets centered at the M point.”

**Comment 2:** What is the length scale that determines if the geometrical extension (edge, corner) is
large enough to be identified as such by the superconductor? And more specific, how certain are
you that the corner you studied with an extension ~ 4 nm is large enough for a Majorana state to
emerge there?

**Response 2:** We thank the reviewer for this critical comment. This is a quite good and important
question. As pointed out by the reviewer, the edge/corner modes in the topological state revealed in
Ref. [30] are localized states near the edge/corner, and the size of these modes (*i.e.*, the decay length

of these modes) are essential for the experimental detections. Actually, the decay length of edge
modes is mainly determined by the bulk superconducting gap, *i.e.*, Δ_{bulk} in Figure R11a, while the
decay length of corner modes depends on the superconducting gap at the edges, *i.e.*, Δ_{edge} in Figure
R11b. More specifically, in such gapped Dirac system, the decay behavior of the localized modes
scales as $e^{-\int dr|\Delta(r)/v_F|}$, where $\Delta(r)$ is the position-dependent superconducting gap and v_F is the
Fermi velocity. If we roughly ignore the location dependence of the superconducting gap, we have
$e^{-\int dr|\Delta(r)/v_F|} \propto e^{-|r/\xi|}$, where $\xi = v_F/\Delta$ is the coherence length.

**Figure R11 | Sketch for the topological edge and corner states in the sign-changed s_{\pm} -wave state.**

Δ_{bulk} stands for the superconducting gap in the bulk material and Δ_{edge} means the superconducting
gap on the (11) and $(1\bar{1})$ edges. Figures are taken from Ref. [30].

For single-layer FeSe/SrTiO₃ system, we have $\Delta_{bulk} \approx 10$ meV and $\xi_{bulk} \approx 3$ nm [Nat. Phys. **11**,
946 (2015)]. As for Δ_{edge} and the corresponding coherence length ξ_{edge} , there is no relevant work
for reference. Nevertheless, since it has been observed that the superconducting gap at the edge is
reduced by less than half compared with that of the bulk (Figures 2c, 3c, S1d and S1e), the ξ_{edge} is
expected to be larger than ξ_{bulk} but not more than one order of magnitude. Below, we discuss the
effects of corner and edge sizes separately:

(1) Corner size——The corner we studied with an extension ~ 4 nm may be not large enough to
stabilize the corner modes. In order to seek a high-quality large size corner, we prepared many
single-layer FeSe/SrTiO₃ films. Figure R12 shows typical STM topographic images of single-layer
FeSe films with different coverage. The size of most topographic images is $400 \text{ nm} \times 400 \text{ nm}$ (scale
816 bar stands for 100 nm). From left to right and from top to bottom, the coverage gradually decreases
(see the mark at the bottom left of each image). In the case of high coverage (Figures R12a-R12o),
there are mainly (01) edges, very few (11) edges, and even fewer (11) and $(1\bar{1})$ intersections due to
the epitaxial growth characteristics of the sample. A typical example is the inset in Figure R12b,

where almost all the edges are along the (01) orientation. In order to increase the probability of (11)
and ($\bar{1}\bar{1}$) intersections, we significantly reduced the coverage (Figures R12p-R12r), expecting some
defects in SrTiO₃ to locally affect the epitaxial growth of FeSe. However, there is still no satisfactory
corner in large size. In addition, due to the low coverage, the connectivity between FeSe islands is
poor (Figures R12p-R12r), resulting in the coupling of the density of states of SrTiO₃ in the tunnel
spectrum. As shown in Figures R12s and R12t, the measured differential conductance on FeSe
decreases rapidly when the voltage deviates from the initial set value, which will make the Majorana
mode difficult to recognize.

Since we cannot obtain large sized corners due to the limitations of molecular beam epitaxy, we
would like to consider what will happen to the Majorana mode when the size is smaller than ξ_{edge} .
In this case, the Majorana modes from different corners can hybridize with each other, causing the
Majorana zero-energy modes to become in-gap bound states distributed symmetrically on both sides
of the zero energy [Sci. Adv. **6**, eaay0443 (2020)]. Notice that the energy of such in-gap bound
states move closer to the superconducting gap edge as the size of the corner becomes smaller. These
in-gap bound states, which can be directly probed by STS, are not observed in Figures 4, S6 and S7,
suggesting that the superconducting states at corners of single-layer FeSe are topologically trivial.

**Figure R12 | STM topographic images ($V_s = 1$ V, $I_t = 50$ pA) of single-layer FeSe/SrTiO₃. a-r,**

Topographic images of single-layer FeSe with various coverage. The coverage of the film has been

indicated in the lower left area of each image. The dimensions of each image are 400 nm × 400 nm

(Figure R12c: 700 nm × 700 nm; Figure R12m: 600 nm × 600 nm; Scale bar: 100 nm). s-t, Typical

tunneling spectra collected on single-layer FeSe films with low (red lines) and high (grey lines)

coverage, respectively. The setup conditions are $V_s = 500$ mV, $I_t = 500$ pA, $V_{\text{mod}} = 5$ mV for s and

$V_s = 50$ mV, $I_t = 500$ pA, $V_{\text{mod}} = 0.5$ mV for t. Due to the coupling of the density of states of SrTiO₃

in the low coverage case, the measured $dI/dV(V)$ is greatly suppressed when $V < 0$.

(2) Edge size——The edges we studied are large enough to stabilize the edge modes. For example,
the topographic image of the (01) edge shown in Figure 3a in the main text is taken from the area
outlined by the blue box in the topographic image shown in Figure R13. It can be seen that the (01)
edge we studied extends 31 nm in space and is 17 nm away from the nearest (01) edge. These
dimensions are much larger than the coherence length ξ_{bulk} . Therefore, if the single-layer
FeSe/SrTiO₃ is in the sing-changed s_{\pm} -wave pairing state, *our STM measurements can detect clear*
*signals for the Dirac edge modes on the (01) edge.*

**Figure R13 | STM topographic images ($V_s = 1$ V, $I_t = 50$ pA) of single-layer FeSe/SrTiO₃.** The
(01) edge shown in Figure 3a in the main text is taken from the area outlined by the blue box.

Based on the discussion above, if the single-layer FeSe/SrTiO₃ is in the sing-changed s_{\pm} -wave
pairing state, one can expect that (i) the spectrum collected at the corner shown in Figure 4 will have
a zero-bias conductance peak or a pair of in-gap bound states located within the two superconducting
peaks and distributed symmetrically about zero bias, and (ii) the spectrum collected at the edge
shown in Figure 3 will have finite and constant differential conductance within the two
superconducting peaks, as illustrated in Figure 1c in the main text. However, in experiments we
detect neither such signatures for the corner modes nor the signatures for the edge modes. Therefore,
we conclude that the pairing symmetry of single-layer FeSe is the sign-preserving s -wave rather
than the sign-changed s_{\pm} -wave.

In the revised version, we have added some discussion of effects of edge/corner size on the detection
of topological modes (Lines 107-111, 132-135, 137-140). In addition, in the revised Supplementary
Information, we have added Supplementary Note 2 to present the detailed discussion described
above.

**Comment 3:** With respect to your numerical study and Fig. S7; in the case with $V_{\{b\}}=6$ [Fig.
$f_{\{1,2\}}$ and $g_{\{1,2\}}$] the increase of the inter-layer hopping decreases the weight of the Majorana
corner state. Yet, with $V_{\{b\}}=5$ [Fig. $f_{\{3,4\}}$ and $g_{\{3,4\}}$] it seems to me that the Majorana state
benefits from the increased inter-layer hopping. Any explanation on that?

**Response 3:** We appreciate the reviewer for such careful review.

In our opinion, the phenomenon pointed out by the reviewer is most probably from the finite-size
effect. To confirm this, we change the lattice size of the model used in Supplementary Note 5 and
calculate the local density of states at the corner in the presence of the substrate. In Supplementary
Note 5, the superconductor layer has 56×56 lattice sites and the substrate layer has 64×64 lattice
sites (Figure S9b). Here, we make the superconductor layer and the substrate layer have the same
lattice size (both of the two layers have 60×60 lattice sites). The results are presented in Figure R14.
It is obvious that the zero-bias peak becomes weaker as the interlayer coupling t_h becomes larger.

**Figure R14 | Local density of states at the corner in the presence of the metallic substrate.** In
the calculations, we consider similar real-space structure with that in Figures S9g1-S9g4 in
Supplementary Note 5; And the only difference is that we make the superconductor layer and the
substrate layer have the same lattice size (both of the two layers have 60×60 lattice sites here). We
set $t_h = 0.2$ in **a**, $t_h = 0.5$ in **b** and $t_h = 0.8$ in **c**, and other parameters are the same with these in
Figure S9g3. For comparison purposes, the vertical coordinates of the Figure here and Figure S9g1-
S9g4 in Supplementary Note 5 are the same.

In fact, for the real-space structure in Figure S9b in Supplementary Note 5, we have checked the
results for larger interlayer coupling ($t_h = 0.8$ and other parameters are all the same with these in
Figure S9g3), and find that the zero-bias peak is much weaker than that in Figures S9g3 and S9g4,
which is consistent with the expectation.

In the revised Supplementary Information, we have explained that the higher zero-bias peak in
Figure S9g4 than in Figure S9g3 is due to the finite-size effect (Lines 189-192 in the Supplementary
Information).

Summary of changes

(The major revisions are also marked up in the revised manuscript/Supplementary Information)

A. Main Text

- (1) We have added Dr Xianxin Wu, who contributed to the numerical calculation, to the list of authors.
- (2) We have explicitly stated that this technique is suitable for the study of pairing symmetry in iron-based superconductors whose Fermi surface consist only of two electron pockets centered at the M point. (Lines 61-64)
- (3) We have changed the expression “second-order superconducting state” to “second-order topological superconducting state” to avoid any ambiguity.
- (4) We have emphasized that we discuss the pairing symmetry of single-layer FeSe from the topological perspective. (Lines 77-78)
- (5) We have given a more detailed description of the quality of the (01) edge. (Lines 105-111)
- (6) We have added some descriptions of the little effects of edge/corner size on detecting topologically non-trivial modes. (Lines 107-111, 132-135, 137-140)
- (7) We have added some discussion that the reduction of gap magnitude has no effect on the detection of topologically non-trivial modes. (Lines 161-169)
- (8) We have added a sentence to illustrate the energy resolution that meets the requirements of this experiment. (Lines 172-173)
- (9) We have added some descriptions of the effects of magnetism on detecting topologically non-trivial modes. (Lines 218-221)
- (10) We have switched the colors of the arrows in Figures 2a and 3a to keep the colors consistent.
- (11) We have added the normalized differential conductance to Figures 2e and 3e to better show how close that is to “zero”.
- (12) We have updated the atomically resolved image of Figure 3a.
- (13) We have carefully polished the main text thoroughly.

B. Supplementary Information

(1) We have re-fitted the tunneling spectra using the Dynes formula without the broadening factor

Γ . Table S1, Figure S1 and Figure S14 have been modified accordingly. (Supplementary Notes 1
and 8)

(2) We have added Supplementary Note 2 to the Supplemental Material to include a discussion of
the effects of edge/corner size on the detection of topological modes. (Supplementary Note 2)

(3) We have added some descriptions of the tunneling spectra in Figures S6 and S7. (Lines 120 and
126 in Supplementary Note 4)

(4) We have added some explanations for the anomalous variation in the height of the zero-bias
peak. (Lines 189-192 in Supplementary Note 5)

(5) We have added Supplementary Note 6 to the Supplemental Material to include a discussion of
the effects of decrease in gap magnitude near the edge on the detection of topological modes.
(Supplementary Note 6)

(6) We have added Supplementary Note 7 to the Supplemental Material in which we show the
sensitivity of our tunneling junction to differential conductance anomalies. (Supplementary Note 7)

Reviewers' Comments:

Reviewer #1:

Remarks to the Author:

I would like to thank the authors for such a clarifying and detailed answer. I now see the flaws in my previous judgement and also that authors have taken a constructive position, which has led to a significant improvement of the manuscript. I fully agree with authors that the state they address is new and that their finding is very relevant, as it contains a neat search for topological features, with clear-cut results.

I think that the manuscript is now easier to read and can be more easily placed in a wide context and have no doubts that the paper should be accepted as is, albeit the point I mention now.

This is a minor point but clearly an error which should be corrected. What authors mention as the "Dynes" formula in the first equation of the supplement is not such a "Dynes" formula. Dynes formula (PRL 41, 1509 and their reference 35) includes lifetime broadening, which is here not used and probably not needed either. It is very confusing to use such a term for the simple temperature smeared BCS density of states (first equation of the supplement). Authors should remove the reference 35 of the main paper and simply state that they use a modified temperature smeared BCS density of states, referencing formula 1 of the supplementary information. Further references to the name of Dynes (which appears many times in the paper as well as in the supplement) should be removed. I say this with all my respect to such a reputed and incredibly productive scientist as Dynes, but I guess that we would all agree that the important insight he and his colleagues won in those papers does not really apply to the context of the present manuscript.

Reviewer #3:

Remarks to the Author:

In the revised manuscript, the authors have taken great care to thoroughly examine the sizes of the geometrical/topological defects. They have convincingly found that the edges studied are large enough to potentially host edge states, which is correspondingly, their strongest argument in the conclusion that topological features are absent.

As for the additional evidence related to the potential corner modes, they have admitted, and clarified that the corners might not be large enough for clear Majorana zero-energy states to appear, yet, they would still expect signatures related to hybridized Majorana modes.

For the latter conclusion, I am still somewhat confused. It is argued that a corner mode hybridizes with another corner mode.

First, what does this mean if there is only one isolated $(11)/(1\bar{1})$ -corner, [maybe connected with a $(11)/(10)$ corners [or kinks]]? One would not expect Majorana states at such kinks, correct?

Second, imagine the following scenario. There is a clear long edge with a tiny tiny spike (corner) [be it one, two, three, etc. atoms only] somewhere along it. One would not expect any meaningful corner-features related with it. How does this limit connect with the hybridized-Majorana story? In other words, if corners are isolated and too small, why would the limiting case not just be the absence of corner-related phenomena?

Author Rebuttals to Comments

We thank the reviewers for their recommendations and constructive comments. We have taken these
comments and suggestions seriously and carried out further numerical simulations and analysis.

With these revisions, we hope that all concerns raised by the reviewers have been well addressed.

Below, please find our response letter and the list of main changes. In the point-by-point response

letter, our replies are in **blue** typeface. The modifications to the manuscript/Supplementary

Information are described or quoted in **red**. A detailed list of changes is attached at the end of this

document.

-----

Response to the Reviewer #1

-----

I would like to thank the authors for such a clarifying and detailed answer. I now see the flaws in
my previous judgement and also that authors have taken a constructive position, which has lead to
a significant improvement of the manuscript. I fully agree with authors that the state they address is
new and that their finding is very relevant, as it contains a neat search for topological features, with
clear-cut results.

I think that the manuscript is now easier to read and can be more easily placed in a wide context and
have no doubts that the paper should be accepted as is, albeit the point I mention now.

**Authors:** We thank the reviewer for the positive remarks on the novelty, relevance, and broad
interest of our work. We also appreciate the reviewer's recommendation for the publication of our
manuscript in *Nature Communications*.

**Comment 1:** This is minor point but clearly an error which should be corrected. What authors
mention as the "Dynes" formula in the first equation of the supplement is no such "Dynes" formula.
Dynes formula (PRL 41, 1509 and their reference 35) includes lifetime broadening, which is here
not used and probably not needed either. It is very confusing to use such a term for the simple
temperature smeared BCS density of states (first equation of the supplement). Authors should
remove the reference 35 of the main paper and simply state that they use a modified temperature
smeared BCS density of states, referencing formula 1 of the supplementary information. Further
references to the name of Dynes (which appears many times in the paper as well as in the supplement)
should be removed. I say this with all my respect to such a reputed and incredibly productive
scientist as Dynes, but I guess that we would all agree that the important insight he and his
colleagues won in those papers does not really apply to the context of the present manuscript.

**Response 1:** We agree with the reviewer that the use of the term "Dynes formula" is inappropriate
in this work. As suggested by the reviewer, we have removed the reference 35 from the main text
and stated that we used a temperature smeared BCS density of states (Lines 97-98 and 117-118 in
the main text, Lines 27-28, 35-36, 368-369, and 380 in the Supplementary Information).

-----

Response to the Reviewer #3

-----

In the revised manuscript, the authors have taken great care to thoroughly examine the sizes of the
geometrical/topological defects. They have convincingly found that the edges studied are large
enough to potentially host edge states, which is correspondingly, their strongest argument in the
conclusion that topological features are absent.

As for the additional evidence related to the potential corner modes, they have admitted, and
clarified that the corners might not be large enough for clear Majorana zero-energy states to appear,
yet, they would still expect signatures related to hybridized Majorana modes.

For the latter conclusion, I am still somewhat confused. It is argued that a corner mode hybridizes
with another corner mode.

**Authors:** We thank the reviewer for the positive evaluation of our revised manuscript. The reviewer
has fully caught the main point of our response of the preceding round; and we really appreciate the
reviewer for the comment, “They have convincingly found that the edges studied are large enough
to potentially host edge states, which is correspondingly, their strongest argument in the conclusion
that topological features are absent”. In the following, we provide the point-by-point response to
address the reviewer’s concerns.

**Comment 1:** First, what does this mean if there is only one isolated $(11)/(1\bar{1})$ -corner, [maybe
connected with a $(11)/(10)$ corners [or kinks]] ? One would not expect Majorana states at such kinks,
correct?

**Response 1:** We thank the reviewer for the insightful and constructive comment. We absolutely
agree with the reviewer on the point that, “if there is only one isolated $(11)/(1\bar{1})$ corner [maybe
connected with a $(11)/(10)$ corners [or kinks]], one would not expect Majorana states at such kinks”.
As pointed out by the reviewer, the existence of the corner Majorana states strongly depends on the
specific condition of the corners (or kinks). Although the theoretical work in Ref. [30] shows that
the topological state is protected by the mirror symmetry, the corner Majorana mode can in fact
exist under more relax conditions. That is, for single-layer FeSe with sign-changed s_{\pm} -wave pairing,

the Majorana Kramers' pair can even exist in the absence of the mirror symmetry as long as the
 mass domain picture in Ref. [30] maintains. The effect of mirror symmetry on corner modes is that
 the classification of the corner Majorana Kramers' pairs is \mathbb{Z}_2 in the absence of mirror symmetry
 and is \mathbb{Z} in the presence of mirror symmetry. Here, we can use two examples shown in Figure R1
 (and the caption for Figure R1) to illustrate the above arguments.

 **Figure R1 | The mass domain picture in Ref. [30].** In **a** and **b**, the dashed lines refer to the edges
 with various orientations such as (11), $(1\bar{1})$ and (01), whose mass terms are marked by $+m$, $-m$
 and 0. For the topological state in FeSe, one Majorana Kramers' pair can exist at the corner between
 edges carrying opposite mass terms. Specifically, in Figure R1a, besides the (11)/ $(1\bar{1})$ corner, the
 Majorana Kramers' pair exists at the corner between the $(1\bar{1})$ edge and the edge located in the angle
 labeled by light blue; In Figure R1b, the Majorana Kramers' pair exists at the corner labeled by light
 gray circles and cannot exist at the corner labeled by the light green circle. Actually, the (01) edge
 corresponds to a critical case, where the mass term vanishes and the related corner Majorana modes
 evolve into the Majorana Dirac cones propagating along the edge.

 For the case raised by the reviewer, we carry out numerical simulations based on the model
 described in the Supplementary Note 5, and present the results in Figure R2. As expected, only the
 (11)/ $(1\bar{1})$ corner hosts a Majorana Kramers' pair and there is no corner Majorana modes at the
 (11)/(01) corner. The (11)/(01) corner is special because the mass term vanishes on the (01) edge,
 and two Majorana Dirac cones exist on the (01) edge. In fact, it can be said that the two Majorana
 Dirac cones existing along the (01) edge are evolved from the corner Majorana modes at the
 (11)/(01) corners. Please note that the corner mentioned here is between the (11) edge and an edge
 slightly off the (01) edge located in the light blue angle indicated in Figure R1a.

**Figure R2 | numerical simulations for corner modes.** We consider the lattice geometry in **a** and
 simulate the corner modes based on the model described in Supplementary Note 5. **b** and **c** show
 the local density of states at the two corners labeled by light gray and light green circles, respectively.
 As expected, the $(11)/(1\bar{1})$ corner host a Majorana Kramers' pair, while there is no well-defined
 Majorana corner mode at the $(11)/(01)$ corner as pointed out in the caption in Figure R1.

Having clarified the electronic states at each corner, a natural question is how does the zero-energy
 mode evolve if the $(11)/(1\bar{1})$ corner is very small, *i.e.*, the (11) and $(1\bar{1})$ edges in Figures R2a and
 R5b are very short? In this limiting case, the zero-energy mode at the corner will hybridize with the
 *gapless edge mode* located at the (01) edge and become a pair of bound states under a strong strength
 of hybridization. Such hybridization is reasonable because the wave functions of corner and edge
 modes overlap in space due to the small size of the corner. Although it is difficult to accurately
 simulate the hybrid corner state, the hybrid corner state should be similar and have the same size
 dependence as that shown in Figure R4.

**In the revised version, we have extended the scope of hybridization from corner and corner modes**
 **to corner and corner/edge modes (Lines 135-136 in the main text, Lines 93-116 in the**
 **Supplementary Information).**

**Comment 2:** Second, imagine the following scenario. There is a clear long edge with a tiny tiny
 spike (corner) [be it one, two, three, etc. atoms only] somewhere along it. One would not expect any
 meaningful corner-features related with it. How does this limit connect with the hybridized-
 Majorana story? In other words, if corners are isolated and too small, why would the limiting case
 not just be the absence of corner-related phenomena?

**Response 2:** We agree with the reviewer's view of point that, in the small limit one would not
 expect any meaningful corner features (especially for the case mentioned by the reviewer, where
 the long edge has a tiny tiny spike/corner).
 However, the condition in our experiment is a little different. As shown in Figure 4 in the main text,
 the corner is embedded in a region where FeSe is missing. To understand the possible corner
 Majorana modes on such a defect, we can first consider a circular defect in the sample, as illustrated
 in Figure R3. In this scenario, if the circular defect is large enough, we can apply the mass domain
 picture described in Ref. [30]. Then, one would expect four Majorana Kramers' pairs on the circular
 defect with each pair located at the sign-change point of the mass term, as indicated by red circles
 in Figure R3. When the circular defect becomes irregular, the Majorana Kramers' pairs would not
 thoroughly disappear. If the defect is small (or two of the sign-change points of the mass term, *i.e.*,
 the red circles in Figure R3, are near with each other), the corner Majorana modes would hybridize
 and become in-gap bound states at finite energy. In addition, the smaller the defect, the stronger
 hybridization of the corner Majorana modes, and the closer the energy of the in-gap bound states is
 to the bulk superconducting gap. We have also carried out numerical simulations based on the model
 described in Supplementary Note 5, and the results shown in Figure R4 are consistent with the above
 analysis.

 **Figure R3 | Schematic diagram of a circular defect, namely a hole (the white region) embedded**
 **in the sample (the gray region).** If the hole is large enough, one would expect four Majorana
 Kramers' pairs on the circular defect with each located at the point where the mass term changes
 sign (the red circle).

It is worth mentioning that in the small defect limit (for instance, the defect is a missing atom in the
 lattice), the condition becomes similar to the impurity problem (the missing atom behaves like a
 nonmagnetic impurity); And the argument is similar to our response to “Comment 3.3” raised by
 reviewer #1 in the previous-round review). In such case, the defect will bound a pair of in-gap states
 in the sign-changed s_{\pm} -wave pairing state, which is also consistent with the above analysis.

**Figure R4 | Effect of size on the hybridization of the corner Majorana modes.** Based on the
 model described in Supplementary Note 5, we simulate the bound states assuming a circular defect
 (a hole) in the lattice. In the simulation, we take periodic boundary condition in the outer edges. **a1-**
 **a3** show the three conditions where the holes have different size, and **b1-b3** show the corresponding
 local density of states on the edges of the holes. As shown, there are always in-gap bound states on
 the edge of the hole. As the hole becomes smaller, the energy of the lowest-energy bound state
 moves to the superconducting gap edge (the bulk has a superconducting gap of about 0.2 as shown
 in Figure S9 in the Supplementary Information).

According to the above analysis and arguments, though the defect in our experiments is not large
 enough, the corner in such a defect can still bound some in-gap states. However, in our STM
 measurements, we did not observe any signals for such in-gap bound states.

**In the revised Supplementary Information, we have enriched the discussion of hybridization of zero-**
 **energy modes (Lines 93-116 in the Supplementary Information).**

Finally, let's go back to the two special cases raised by the reviewer as shown in Figure R5. In
 Figure R5a, two $(11)/(1\bar{1})$ corners separated by only a few atoms are located on a clear long (11)
 edge. Similar to the Figures R4a1 and R4b1, the corner mode denoted by the red circles in Figure
 R5a will hybridize strongly and evolve into bound states very close to the superconducting
 coherence peak, which is difficult to observe experimentally. In Figure R5b, although the corner is
 isolated, the corner mode can be hybridized with the *gapless edge mode* along the (01) edge and
 become in-gap bound states as well. Therefore, we argue that there is no meaningful corner-features
 at a tiny tiny spike (corner).

**Figure R5 | Sketch for a clear long edge with a tiny tiny spike (corner) [be it one, two, three,**
 **etc. atoms only] somewhere along it. The green lines in b denote the gapless edge states located**
 **at the (01) edge.**

Summary of changes

(The major revisions are also marked up in the revised manuscript/Supplementary Information)

(1) We have removed the reference 35 from the main text and stated that we used a temperature
smeared BCS density of states (Lines 97-98 and 117-118 in the main text, Lines 27-28, 35-36, 368-
369, and 380 in the Supplementary Information).

(2) We have extended the scope of hybridization from corner and corner modes to corner and
corner/edge modes, and discussed in detail the effect of corner size on the hybridization of corner
Majorana modes (Lines 135-136 in the main text, Lines 93-116 in the Supplementary Information).

(3) The author's affiliation has been updated.

(4) The grant number of the Fund 51788104 has been changed to 52388201 (Line 247 in the main
text).

REVIEWERS' COMMENTS

Reviewer #1 (Remarks to the Author):

The manuscript has been further enriched by discussions on possible features regarding majorana modes at edges and corners. Authors have convincingly addressed all issues raised by Referees. I strongly support publication of this work.

Reviewer #3 (Remarks to the Author):

I thank the authors for the extra effort taken into further elucidating the nature of hybridized corner modes, and I appreciate the insightful explanations. I have no further questions/doubts, and I fully recommend publication of the manuscript---as is--- in Nature Communications.

Author Rebuttals to Comments

Response to the Reviewer #1

The manuscript has been further enriched by discussions on possible features regarding majorana modes at edges and corners. Authors have convincingly addressed all issues raised by Referees. I strongly support publication of this work.

Authors: We thank the reviewer for the constructive suggestions and comments on our work, which have helped to take this work to a new level.

Response to the Reviewer #3

I thank the authors for the extra effort taken into further elucidating the nature of hybridized corner modes, and I appreciate the insightful explanations.

I have no further questions/doubts, and I fully recommend publication of the manuscript---as is--- in Nature Communications.

Authors: We thank the reviewer's recommendation for the publication of our manuscript in Nature Communications.